# Tomographic near-eye displays

Seungjae Lee [1], Youngjin Jo[1], Dongheon Yoo[1], Jaebum Cho[1], Dukho Lee[1] & Byoungho Lee [1]

The ultimate 3D displays should provide both psychological and physiological cues for depth recognition. However, it has been challenging to satisfy the essential features without making sacrifices in the resolution, frame rate, and eye box. Here, we present a tomographic near-eye display that supports a wide depth of field, quasi-continuous accommodation, omni-directional motion parallax, preserved resolution, full frame, and moderate field of view within a sufficient eye box. The tomographic display consists of focus-tunable optics, a display panel, and a fast spatially adjustable backlight. The synchronization of the focus-tunable optics and the backlight enables the display panel to express the depth information. We implement a benchtop prototype near-eye display, which is the most promising application of tomographic displays. We conclude with a detailed analysis and thorough discussion of the display's optimal volumetric reconstruction. of tomographic displays.

[1] School of Electrical and Computer Engineering, Seoul National University, Gwanak-Gu Gwanakro 1, Seoul 08826, Republic of Korea. Correspondence and requests for materials should be addressed to B.L. (email: byoungho@snu.ac.kr)

An ideal three-dimensional (3D) display system provides an immersive and realistic experience. To approach the ideal 3D display system, the following points should be considered: first, the visual characteristics of a real object that make it look real; second, alleviation of the artificiality that comes from state-of-the-art displays; finally, those essential features must be realized without sacrificing figures of merit, such as the resolution, frame rate, and eye box. The human visual system understands the real world and perceives depth information of 3D objects via psychological and physiological cues. Psychological cues are related to the visual effects that are usually observed in daily life, including shading, perspective, illumination, and occlusion. With advancements in 3D rendering and computer graphics, these psychological cues can be reproduced via ordinary two-dimensional (2D) display panels. On the other hand, physiological cues refer to the physical states of the two eyes and objects in terms of convergence, accommodation, and motion parallax. To induce physiological cues, a specific display system is required, such as light field displays[1–7], stereoscopes with focus cues[8–15], and holographic displays[16–20].

Several display technologies have been introduced and studied to reconstruct physiological cues. However, it has been challenging to provide all of the convergence, accommodation, and motion parallax without sacrificing the display performance. Mostly, the ability of the display system to reproduce physiological cues involves sacrificing the resolution[1,5,6,8,10,14,15,21], frame rate[2,9,11,12], viewing window[16,20], or eye box[13,17,19]. For instance, light field displays with focus cues suffer from a trade-off among the spatial resolution, angular resolution, depth of field[22], and frame rate. Several optical systems have been introduced to reconstruct four-dimensional light fields using lens arrays[1], multi-projections[2,6], or layered structures[3,5–7]. Although each approach has distinct advantages, they share a common limitation (i.e., trade-offs) that comes from the large amount of information required for the reconstruction of light fields. Recently, holographic displays have been spotlighted, especially in near-eye display applications, as an alternative approach to light field displays. Several holographic near-eye display prototypes were analyzed thoroughly regarding enhancements in form factor[17,20] and tolerance[23]. Nevertheless, holographic displays suffer from a trade-off between the field of view and exit pupil, which is related to the limited bandwidth of spatial light modulators. In addition, holographic displays possess challenging issues, such as speckle noise and large computational demand.

Multi-plane displays are also able to provide users with depth information by floating multiple discrete focal planes. In multi-plane displays, it is important to achieve a large number of focal planes for high resolution, wide depth of field, and continuous accommodation cues. However, it is challenging to increase the number of focal planes without declining the frame rate or form factor. The most feasible approach for increasing the focal plane number is to employ temporal multiplexing[9,11,12,15,21] with synchronization of a display module and focus-tunable optics. A state-of-the-art display, which is refreshed at 240 Hz, may optimally reconstruct four focal planes by sacrificing the frame rate. However, it is difficult to cover a wide depth of field while providing continuous focus cues. Recently, Chang et al.[24] and Rathinavel et al.[25] reported volumetric displays that have a large number of focal depths, i.e., 40 and 280, respectively. The display module was implemented using a digital micromirror device (DMD) and a high-dynamic range illumination source using a light-emitting diode (HDR LED). The DMD and the HDR LED are synchronized for direct digital synthesis[26] that decomposes color images into binary image sequences. Applying direct digital synthesis, the prototypes of two research groups result in an efficient binary representation of 3D imagery. However, both prototypes have some disadvantages in terms of frame rate or bit depth. Chang et al.'s prototype[24] lacks bit depth and frame rate because it reproduces 8-bit gray images updated at 40 Hz. Rathinavel et al.'s prototype[25] also suffers from the limited bit depth because each focal plane image has a binary bit depth.

Here, we present a tomographic near-eye display that effectively alleviates the trade-off among the spatial resolution, depth of field, and frame rate. The core idea of tomographic displays is the synchronization of focus-tunable optics (e.g., focus-tunable lens or motorized stage with a lens) and a fast spatially adjustable backlight (FSAB). The combination of the two elements allows an ordinary display panel to express depth information without the loss of resolution or frame rate. As a result, our benchtop prototype is shown to support a wide depth of field (from 0.18 m to infinity), quasi-continuous accommodation (80 planes), omnidirectional motion parallax, original display panel resolution (450 × 450), full frame (60 Hz), and moderate field of view (30°) within a sufficient exit pupil (7.5 mm). We also present a detailed analysis and valuable discussion for the advanced applications of tomographic displays. First, we introduce the optimized rendering method for focal plane images in consideration of the occlusion boundaries and the accurate focus cues. Second, tomographic displays are expected to alleviate the specific optical aberration (i.e., field curvature) with a modification of the rendering. Third, high-dynamic range (HDR) displays[27,28] are shown to be a feasible application of tomographic displays. We conclude with a discussion about some limitations of tomographic displays such as brightness, form factor, and computational load, which would open new research fields.

## Results

**Tomographic near-eye displays for virtual reality.** Figure 1 illustrates the procedure for the reconstruction of 3D volumetric objects via tomographic displays. Note that the focus-tunable optics are represented by a focus-tunable lens for intuitive illustration. In the single cycle, the focus-tunable lens modulates the focal length, so that the images of the pixels scan along the specific range of the depth. At the same time, the FSAB determines the depth information of each pixel via illumination at the appropriate moment. The FSAB projects a binary image sequence onto the display panel, which is synchronized with the focus-tunable lens. The binary image sequence is derived from the depth information of the 3D volumetric objects. In summary, while the display panel performs the common role to reproduce a 2D image that includes color and gradation, the focus-tunable lens and the FSAB enable the 2D image to have depth information.

The display module, which consists of the FSAB and the display panel, performs the key role in generating a large number of focal planes at 60 Hz. Because only binary images are handled, it is feasible to drastically increase the frame rate of the FSAB. Binary images could be updated at an extremely high frame rate, which could not be achieved by state-of-the-art display panels (<240 Hz). On the other hand, the display panel includes color and gradation information that the FSAB cannot deal with. In addition, the substantive resolution of a synthesized image is determined by the display panel, which means that the FSAB does not necessarily have a high resolution. In summary, the FSAB and the display panels have complementary relations for 3D displays: resolution, color, and gradation are determined by the display panel, while high frame rate is supported by the FSAB.

We implement a benchtop prototype for tomographic near-eye displays, which can be applied for virtual reality that provides an immersive and comfortable experience. In detail, the prototype is divided into four parts: the DMD projection system for the FSAB,

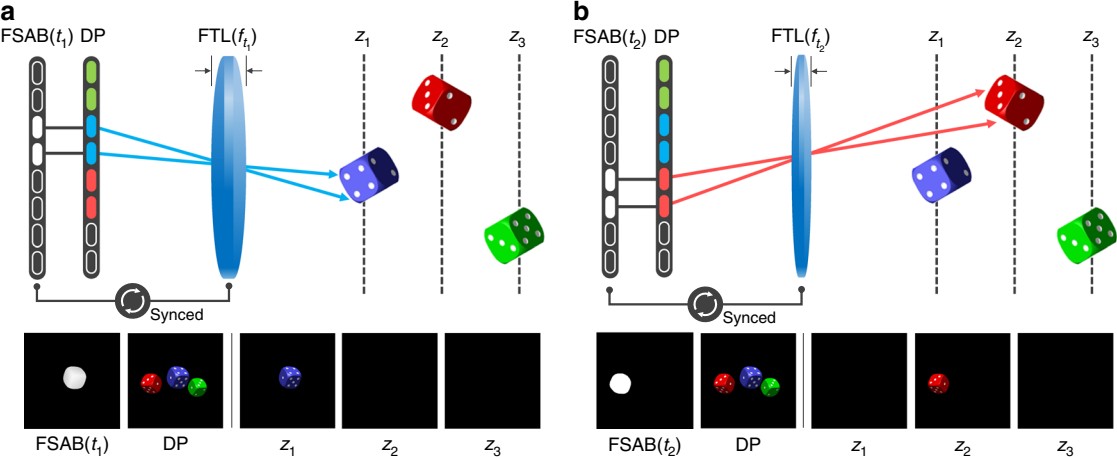

**Fig. 1** Schematic diagram of the principle of tomographic displays. As shown in the figure, the FSAB and focus-tunable lens (FTL) are synchronized to provide users with depth information. **a** When the image of the display panel (DP) is formed at the depth of $z_1$, the FSAB selectively illuminates the blue dice of the display panel. **b** When the focus-tunable lens forms the display panel image at the depth of $z_2$, the FSAB selectively illuminates the red dice of the display panel. As a result, the blue and red dice appear to float at depths of $z_1$ and $z_2$, respectively. Note that each depth can take on a negative value, which means the image can be created at the left side of the focus-tunable lens

the liquid crystal display (LCD) module, the focus-tunable lens, and the eyepiece. The DMD projection system consists of an LED light source with a collimating lens, a total internal reflection (TIR) prism, a DMD, and relay optics with the magnification. The binary image at the DMD is magnified and projected onto the LCD module, which corresponds to the FSAB. Note that the DMD could update the binary images more than 100 times during 1/60 s. The eyepiece secures enough eye relief[22] (50 mm) for the observer while maintaining the optimal field of view that can be achieved with the focus-tunable lens.

Figure 2 demonstrates the display results of the prototype. We employ 3D contents[29] that may show significant variation in depth. 2D projected images and depth maps are used to generate focal plane images. As we can see in the figure, the depth information of the 3D content is well reconstructed via tomographic near-eye displays. The tomographic near-eye display supports the original resolution of the display panel with full color expression. Eighty focal plane images are floated between 5.5D and 0.0D, so that each plane is separated by 0.07D. This separation is narrow enough to provide users with quasi-continuous focus cues[30]. We may observe clear focus cues and blur effects of reconstructed 3D content. Note that motion parallax within the exit pupil is also achieved via tomographic near-eye displays as shown in the figure (see Supplementary Movie 1). The prototype provides a diagonal field of view of 30° within the exit pupil of 7.5 mm. These specifications are verified using the optical simulation tool Zemax and the experiment, which is described in Supplementary Note 6.

Along with the promising display performance demonstrated in the experiment, tomographic near-eye displays have two more advantages. First, they are capable of inserting black frames without decreasing the focal plane number, because our prototype does not necessarily increase the number of focal planes to the utmost limit of the DMD system (~280 planes[25]). We note that black frames contribute to mitigating undesired artifacts when a video is played. Without black frames, users may observe irregular striped patterns caused by simultaneous observations of focal plane image stacks in adjacent frames. Second, we can use an LED array instead of the DMD to implement wearable prototypes. The LED array supports a much lower resolution (8 × 8) and frame rate (<1 kHz) than that of the DMD. In tomographic displays, however, the additional display panel complements the limitation

of the LED array by supporting a much higher resolution (491 dpi) as well as 24-bit depth colors. Supplementary Note 1 presents detailed demonstrations of the necessity of black frames, and Supplementary Note 6 demonstrates wearable tomographic near-eye displays using an LED array.

**Occlusion blending to alleviate depth discontinuity artifact.** Although tomographic displays have various advantages as demonstrated in the previous section, it may be premature to consider tomographic displays as the most promising system for virtual reality. Because focal plane images are merged via addition, tomographic displays could be vulnerable to depth discontinuity artifacts at occlusion boundaries as demonstrated in Supplementary Note 3. Without an adequate solution, the synthesis of focal plane images seems to be artificial when the depth discontinuities are significant. In previous studies related to multi-plane displays, it has been verified that linear[8] or optimal blending[15] could alleviate the depth discontinuities. Unfortunately, tomographic displays could not apply those blending methods directly, because all focal plane images are correlated with each other. Each focal plane image of tomographic displays cannot be determined independently, because the FSAB divides a constant RGB image into multiple focal plane images. Therefore, we need to conceive of an alternative blending method to minimize depth discontinuity artifacts.

In this study, occlusion blending is introduced for tomographic displays, and this approach adopts and combines the ideas of light field synthesis[31] and optimal blending[15]. Although it demands large computation power that hinders real-time operation, this method could significantly minimize the artificial effect that comes from large depth discontinuities. To find optimal focal plane images that satisfy the unique constraint of tomographic displays, we must solve the binary least-squares problem categorized as nondeterministic polynomial-time hardness (NP-hard). Here, we solve the relaxation of the NP-hard problem to verify the ability of tomographic displays to minimize depth discontinuity artifacts. Figure 3 demonstrates the display results of tomographic displays when applying the optimal solution of the binary image sequence and the RGB display image.

**Evaluation of display capability.** To assess tomographic displays, we define two evaluation criteria: upper-bound amplitude and bit

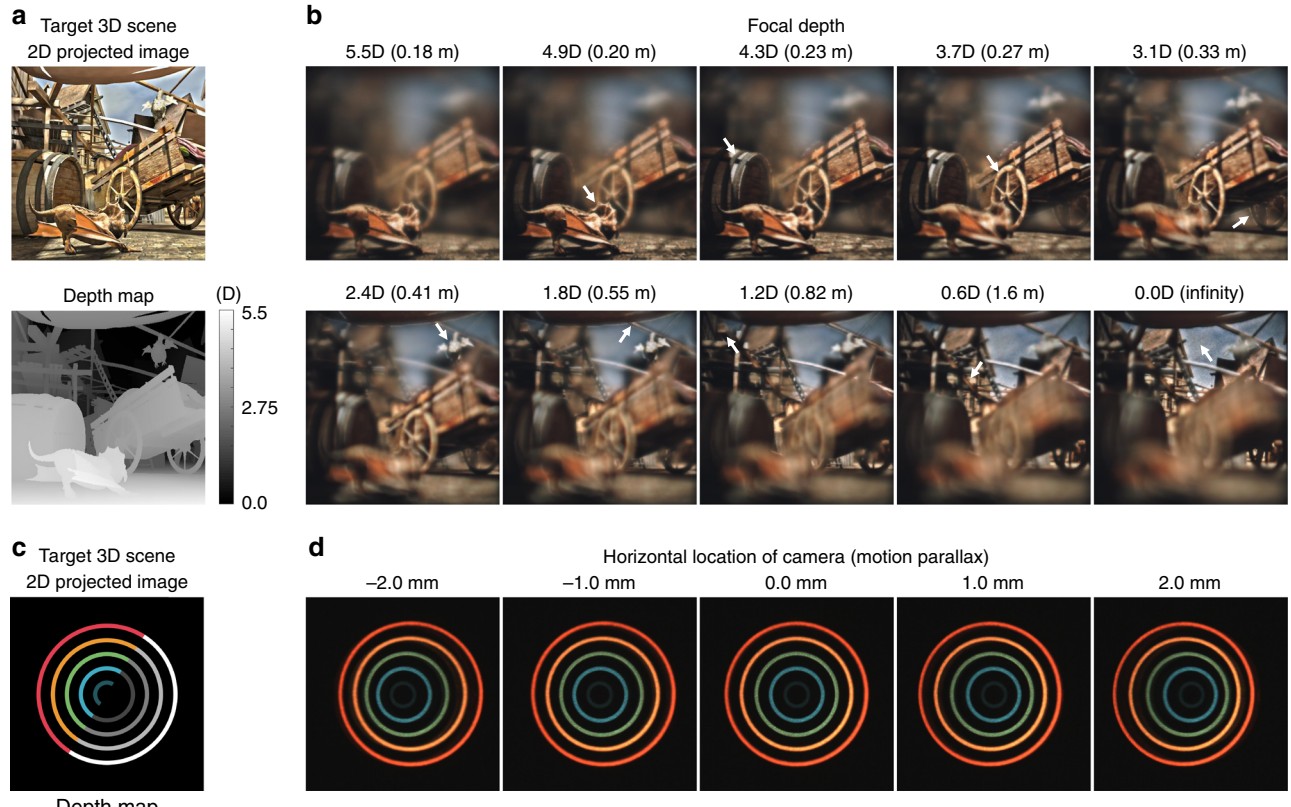

**Fig. 2** Experimental results of tomographic near-eye displays. **a** 2D projected images and the corresponding depth maps are illustrated. The source of 3D content is from the work of Burtler et al.[29]. **b** Experimental results are demonstrated by ten photographs with different focal depths from a CCD camera. As we can see in the photographs, tomographic near-eye displays may support quasi-continuous focus cues (white arrows) while preserving high resolution and contrast. **c, d** We demonstrate a brief experiment to show the motion parallax provided by tomographic near-eye displays. Additional results are available in Supplementary Note 6 and Supplementary Movie 1

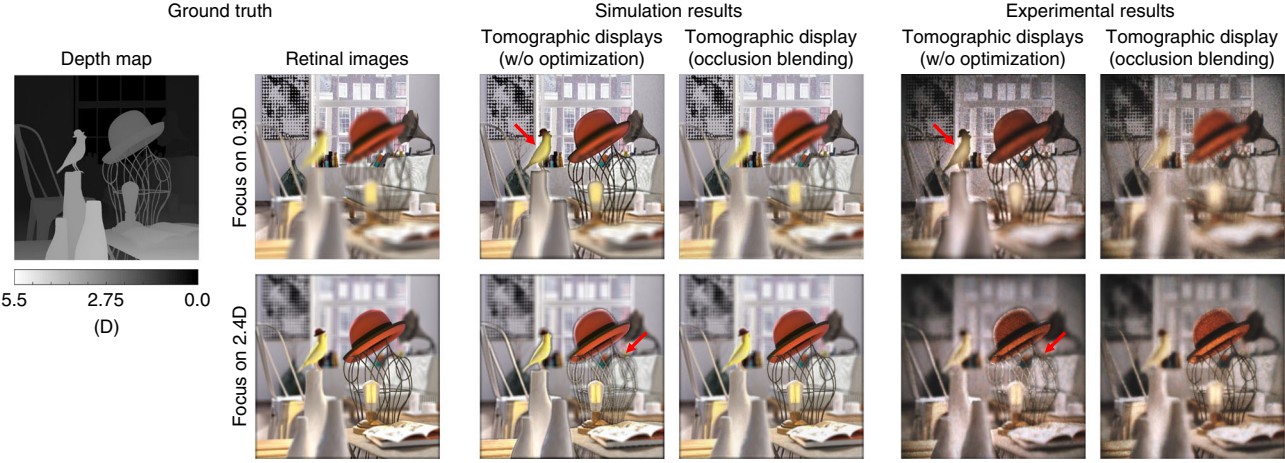

**Fig. 3** Simulation and experimental results demonstrating occlusion blending. A volumetric scene (Source image courtesy: "Interior Scene", www.cgtrader.com) extends along the depth range between 0.0D and 4.0D. As shown in the figure, occlusion blending enables tomographic displays to represent volumetric scenes without noticeable artifacts even at the occlusion boundary (red arrows). Additional experimental results are available in Supplementary Note 2

depth. The upper-bound amplitude is the Fourier coefficient of synthesized retinal images, and bit depth denotes the degree of freedom to modulate pixel brightness. These two criteria provide insights into the contrast, resolution limit, and bit depth of tomographic displays. Figure 4 demonstrates the analysis of the upper-bound amplitude and bit depth supported by tomographic displays. Other state-of-the-art prototypes[24,25] are assessed for

comparison with tomographic displays. Among the candidates, tomographic displays have the most promising potential for the representation of high-frequency information as well as high-dynamic range images. A detailed description of the upper-bound amplitude and bit depth is presented in Supplementary Note 1.

For a more quantitative evaluation of tomographic displays, we also conducted retinal image simulations to analyze how focal

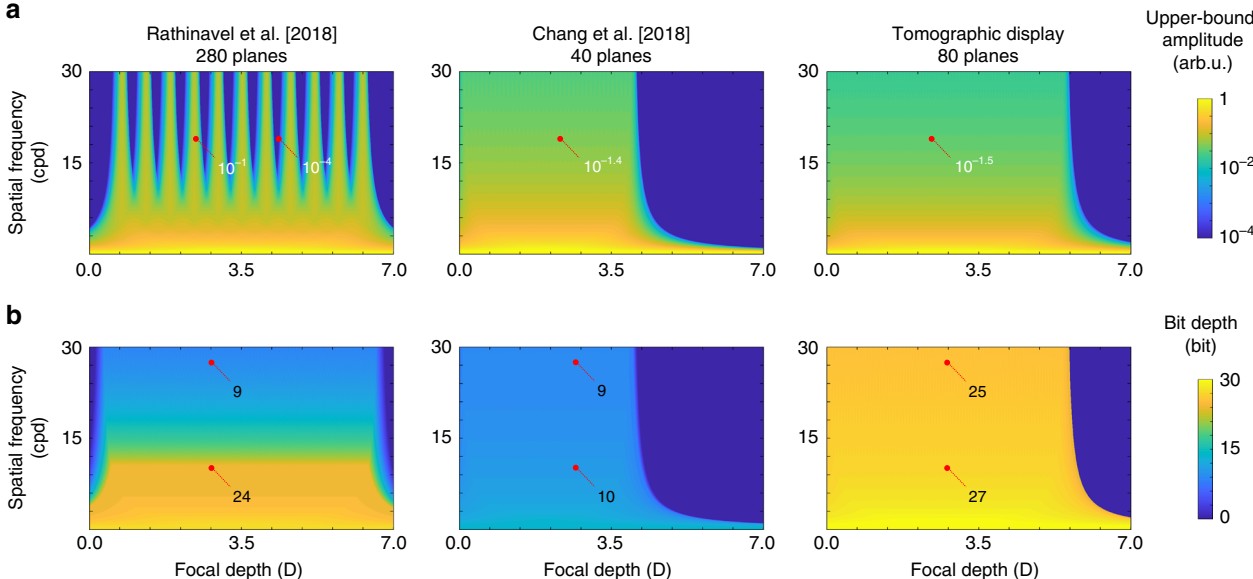

**Fig. 4** Analysis of upper-bound amplitude and bit depth. **a** The upper-bound amplitude is the normalized Fourier coefficient of synthesized retinal images, and **b** the bit depth denotes the degree of freedom in the brightness modulation. Each criterion is plotted according to the spatial frequency of retinal images and the focal depth of observers. The simulation result compares the three prototypes of Rathinavel et al.[25], Chang et al.[24], and this work. Rathinavel et al.'s prototype[25] supports a limited depth of field and bit depth at a high spatial frequency, and Chang et al.'s prototype[24] lacks the bit depth for a full color representation. On the other hand, tomographic displays show reliable performance, regardless of the spatial frequency and focal depth. The exact values at the red points are demonstrated for a precise comparison among those candidates

plane images are synthesized. We compared tomographic displays with 80-plane displays to investigate the drawbacks generated from the slow frame rate of the display panel (60 Hz) in tomographic displays. Contrary to tomographic displays, each focal plane image of 80-plane displays can be independently determined according to the blending method, such as linear blending[8] or optimal blending[15,31]. In this simulation, we assumed that all systems had a resolution limit of 20 cpd, where the horizontal field of view was set to 10°. We employed several visual metrics, including the peak signal-to-noise ratio (PSNR), image quality factor (Q), and HDR-VDP-2[32] that estimates the probability of users being able to detect artifacts. Figure 5 demonstrates the simulation results that verify the validity of using the display panel to increase the number of focal planes and bit depth simultaneously. As shown in the figure, tomographic displays show comparable display performance to that of the 80-plane displays, where each focal plane is determined independently.

**Illumination strategy for real-time operation**. Despite the several merits involved in occlusion blending, such a blending method could be impractical in the real environment, because of the large computational demands. For some applications that require real-time operation, a computationally efficient blending method can be preferred over the accurate representation of occlusion boundaries. In this condition, we may render binary backlight images according to the depth information of 3D scenes. The rendering rule of binary backlight images is determined by the illumination strategy of the display pixels. If we apply this rendering methodology, it is feasible to operate tomographic near-eye displays in real time.

In this study, we optimize the illumination strategy to ensure adequate display performance in terms of brightness, contrast, resolution, and accuracy of focus cues. In the optimization, we consider various requisites of display systems for a comfortable and immersive experience. First, we suppose the lower bound for the illumination time to provide users with adequate brightness of

the synthesized images. The lower bound is considered as a constraint in the optimization. Second, we consider the offset luminance to be a leakage of the backlight source caused by the multiple scattering of light through the backlight diffuser. When the offset luminance is not 0, each pixel of the focal plane is illuminated by a constant brightness, even if the corresponding backlight pixel is turned off. Third, the cost function for the optimization is derived in the frequency domain to reflect human visual characteristics. The most sensitive region for human vision is the spatial frequencies from 4 to 8 cpd[33].

Figure 6 illustrates two illumination strategies: primitive and optimal approaches. In the primitive strategy, each pixel is illuminated by the minimized time when its image is formed at the desired depth. On the other hand, the optimal strategy employs a specific backlight operation that minimizes the cost function described above. If there is no offset luminance ($c = 0$) and a lower bound of brightness ($A_{low} = 0$), the primitive and optimal strategies are identical. When the lower bound of brightness is determined as $A = 0.625m$, the primitive solution is to illuminate the pixel around the desired depth. However, the optimal solution has several lobes for the backlight operation to exploit higher-order intensity distribution. When the offset luminance of the display system is determined as $c = 0.025$, the display system should have enough brightness to surpass the offset luminance. In this condition, the optimal solution may have a longer illumination time than that of the lower bound, as shown in the figure. Compared with the primitive strategy, the optimal strategy enables tomographic displays to have a higher contrast with a sharper peak, so that users can accommodate the desired depth.

**Advanced applications of tomographic displays**. By virtue of the remarkable capability to modulate the depth of imaged pixels, tomographic displays could have various advanced optical applications. For instance, tomographic displays can correct optical aberrations, such as the field curvature, which is usually

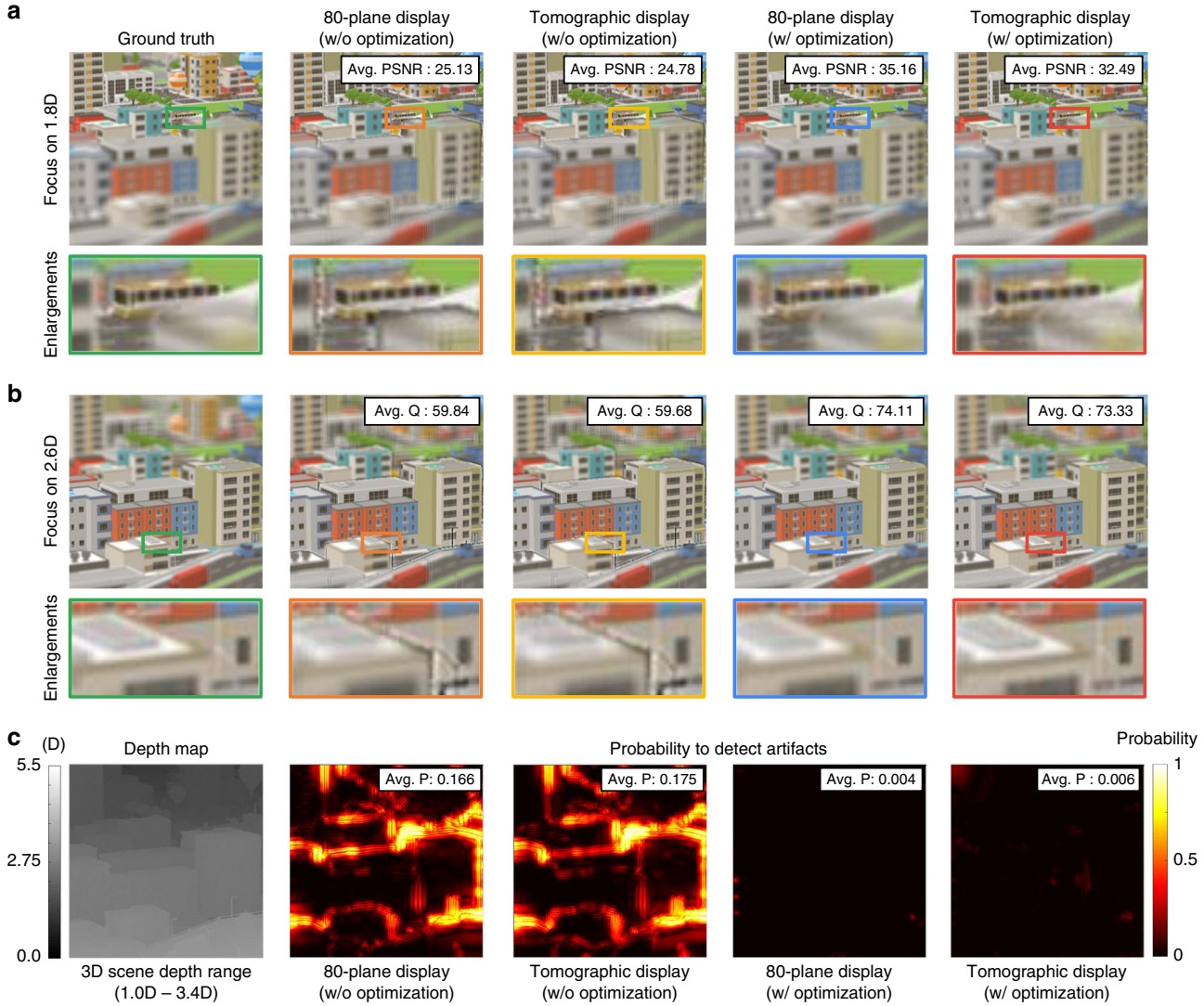

**Fig. 5** Quantitative evaluation of tomographic displays. A volumetric scene (Source image courtesy: "SimplePoly Urban", www.cgtrader.com) extends along the depth range between 1.0D and 3.4D. **a, b** Tomographic displays show the similar display performance with 80-plane displays in terms of average PSNR (Avg. PSNR) and average Q (Avg. Q). The Avg. PSNR is derived from the weighted sum of errors between the ground truth and synthesized retinal images. The weight is estimated by the reciprocal of the optical blur kernel size. The Avg. Q is the mean value of all focal stack images' Q values. **c** We demonstrate a probability map of detection for visual differences between the reconstructed scenes and the ground truth. Each pixel value indicates the weighted average probability over all the focal depths of observers. The average probability (Avg. P) is the mean value of all pixels. In Supplementary Note 4, we present focal plane images for each display system and additional comparison results from other related systems

observed in near-eye display systems[34]. A high-dynamic range (HDR) display[27] is also a feasible application, because the intensity of the backlight could be spatially modulated. We can render HDR focal plane images via modification of the illumination time according to the degree of brightness. In summary, tomographic displays have several advanced applications that provide a more immersive experience.

Figure 7 shows the simulation results that validate the proposed advanced applications of tomographic displays. In the HDR application, the illumination time of each pixel varies according to the desired intensity. The variation range of the illumination time lies between a 0.5× and a 1.5× ratio of the optimal illumination time. Second, a depth map of a 3D scene is pre-compensated to alleviate the optical aberration (i.e., field curvature) of the display system. The pre-compensation is determined by the degree of the optical aberration. Note that we use Seidel coefficients for the simulation of the field curvature.

## Discussion

In this study, we have implemented and explored the prototype for tomographic near-eye displays, which are considered one of the most promising systems to reconstruct 3D objects with continuous focus cues. There are some interesting issues and challenges related to improving performance of tomographic displays, and these issues will provide valuable topics for future research. First, tomographic displays have a limitation in that they have difficulty representing independent focal plane images. Due to limitations, tomographic displays could suffer from artifacts at the occlusion boundary. Although we have demonstrated that tomographic displays are also able to mitigate artifacts via occlusion blending, further research is needed for real-time operation. We believe that the real-time operation could be feasible via a deep learning[35]. Second, enhancing the brightness is important if we aim to stack a large number of focal planes. We could set a lower bound $A_{low}$ to achieve a higher brightness or

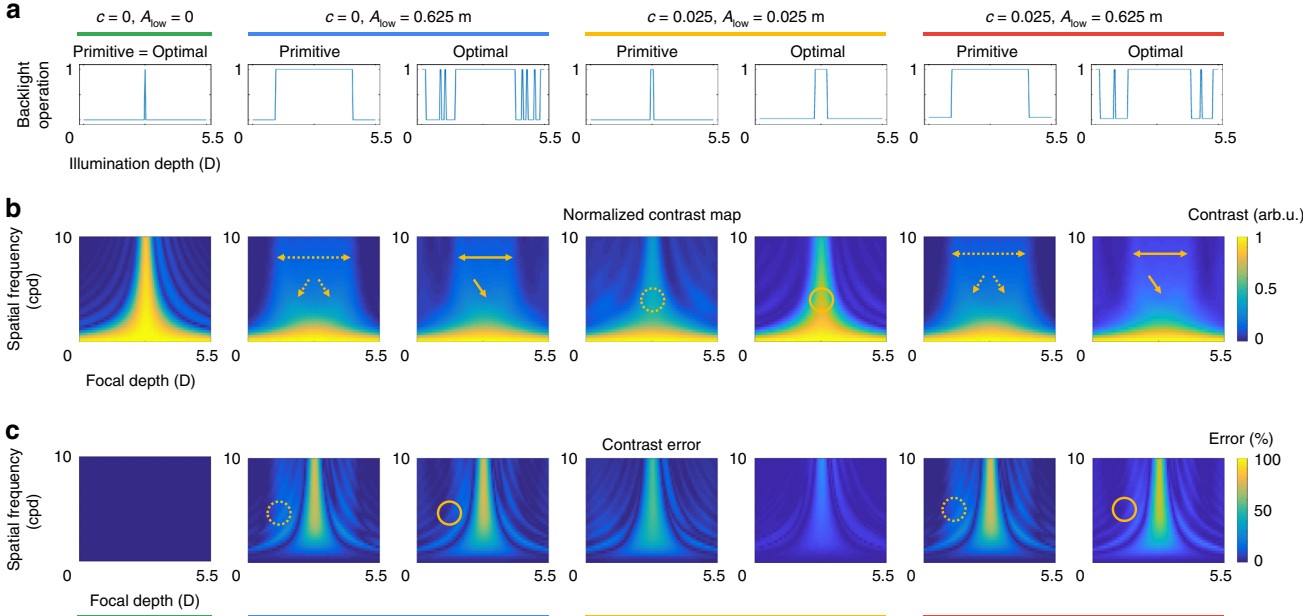

**Fig. 6** Description of optimal illumination strategy. We simulate different conditions according to the degree of offset luminance and desired brightness: ($c = 0$, $A_{low} = 0$), ($c = 0$, $A_{low} = 0.625m$), ($c = 0.025$, $A_{low} = 0.025m$), and ($c = 0.025$, $A_{low} = 0.625m$). **a** We show the backlight operations according to the strategies and circumstances. **b** We demonstrate normalized contrast maps that are achieved by applying a corresponding backlight operation. Normalized contrast maps indicate the relative intensity of retinal images according to the spatial frequency and focal depths. **c** We provide contrast errors determined by the difference between the target and the reconstructed contrast maps. The differences between primitive and optimal strategies are highlighted by dotted or solid arrows and circles. An optimal illumination strategy allows more definite and precise contrast curves. Note that our prototype supposes the third circumstance: ($c = 0.025$, $A_{low} = 0.025m$). Additional results are available in Supplementary Note 6

optimize the optical path of the system to minimize the loss of illumination.

Despite the issues and challenges described above, we believe that tomographic displays could be an effective solution for realizing ultimate 3D displays. In addition, tomographic displays could be modified and adopted for several applications, including tabletop 3D displays[4,19], see-through near-eye displays[36,37], and vision-assistant displays[38]. For tabletop displays, the FSAB could be implemented by using an LED array, and a liquid crystal plate with a lens array could be employed for the focus-tunable optics. For see-through near-eye displays, the eyepiece of the prototype could be replaced by a free-form light guide to combine real-world scenes. For vision-assistant displays, we could use a combination of light field displays, FSAB, and focus-tunable optics. In summary, tomographic displays will be a competitive and economic solution for 3D technologies that aim to provide an immersive experience.

## Methods

**Detailed specifications of the benchtop prototype**. Figure 8 illustrates a schematic diagram of the tomographic near-eye displays. For implementation of the FSAB, a 1.25-inch ultrabright LED spotlight from Advanced Illumination was used as the illumination source. The DLP9500 model from Texas Instruments was employed as the DMD, which supports a full HD resolution (1920 × 1080) and a 0.95-inch diagonal screen. The light-guiding prism for the DMD was customized to satisfy our specifications and convenience for experiments. We also designed relay optics that project the real image of the DMD onto the display screen with a magnification of ×2 . Note that an LED array backlight is also a feasible candidate for the FSAB, which has advantages in the form factor compared with the DMD projection module. In Supplementary Note 6, we demonstrate detailed applications using the LED array as the FSAB.

For the liquid crystal panel, we used the Topfoison TF60010A model, whose backlight module is eliminated. This panel may support high-resolution images of 491 DPI. Note that the partial area (23 × 23 mm) of the LC panel was employed as the display module, due to the limited numerical aperture of the focus-tunable lens and eyepiece. The focus-tunable lens (EL10-30-TC-VIS-12D) of Optotune was selected for the focus-tunable optics, which provides the wide focus-tuning range between 8.3D and 20D for the real-time operation (60 Hz). Using this lens,

tomographic near-eye displays may have a depth of field between 10.5D (0.095 m) and 0D (infinity), while at present, the prototype supports only 5.5D due to the camera lens specifications (TUSS LYM1614, focal length of 16 mm, F number of 1.4, and entrance pupil size of 35.5 mm). Note that we appended the eyepiece module for the prototype to retain enough eye relief (50 mm) for photographs. The eyepiece module consists of two identical camera lenses (Canon EF 50 mm f/1.8 STM) for the 4 f relay system.

**Synchronization of focus-tunable lens and backlight**. As demonstrated in the previous section, we used a DMD module and a focus-tunable lens for the FSAB and focus-tunable optics, respectively. For the synchronization of these two modules, a Data Acquisition (DAQ) board from National Instruments was used to generate the reference clock signals. There are two different signals generated by the DAQ board: one for the focus-tunable lens and the other for the DMD. These two different signals were synchronized using LabView. For the focus-tunable lens, the triangle wave at 60 Hz is generated to modulate the focal length. For the DMD, the square wave at 60 × 80 Hz is generated to update the sequential backlight images on the DMD. A more intuitive description of the synchronization is presented in Supplementary Note 6.

**Least-squares problem for occlusion blending**. To formulate the least-squares problem for occlusion blending, we suppose a tomographic display that supports $m$ focal planes with $n \times n$ resolution. The backlight image sequence consists of $m$ binary images ($\mathbf{b}_1, \mathbf{b}_2, \ldots, \mathbf{b}_m$), each of which has the resolution of $n \times n$. The RGB image can be considered as three grayscale images ($\mathbf{D} = [\mathbf{d}_r, \mathbf{d}_g, \mathbf{d}_b]$) with $n \times n$ resolution. Accordingly, $m$ focal plane images ($\mathbf{L}_1, \mathbf{L}_2, \ldots, \mathbf{L}_m$) are given by the multiplication of the backlight image sequence and RGB image as follows:

$$\mathbf{L}_k = [\mathbf{b}_k, \mathbf{b}_k, \mathbf{b}_k] \odot [\mathbf{d}_r, \mathbf{d}_g, \mathbf{d}_b] \; (k = 1, 2, \ldots, m), \tag{1}$$

where $\odot$ is the Hadamard product of matrices. Then, the derivation of the least-squares problem for optimization is very similar to that of previous studies[10,15,31].

In this study, we apply a multi-view-based optimization algorithm[9,31] that has merits in terms of its computation memory and speed. We note that this approach shows similar performance to that of retinal optimization[15], if we set the eye box size as the pupil diameter. The optimization target is a set of perspective view images ($\mathbf{V}_1, \mathbf{V}_2, \ldots, \mathbf{V}_{p^2}$) within the pupil. In summary, we should solve the

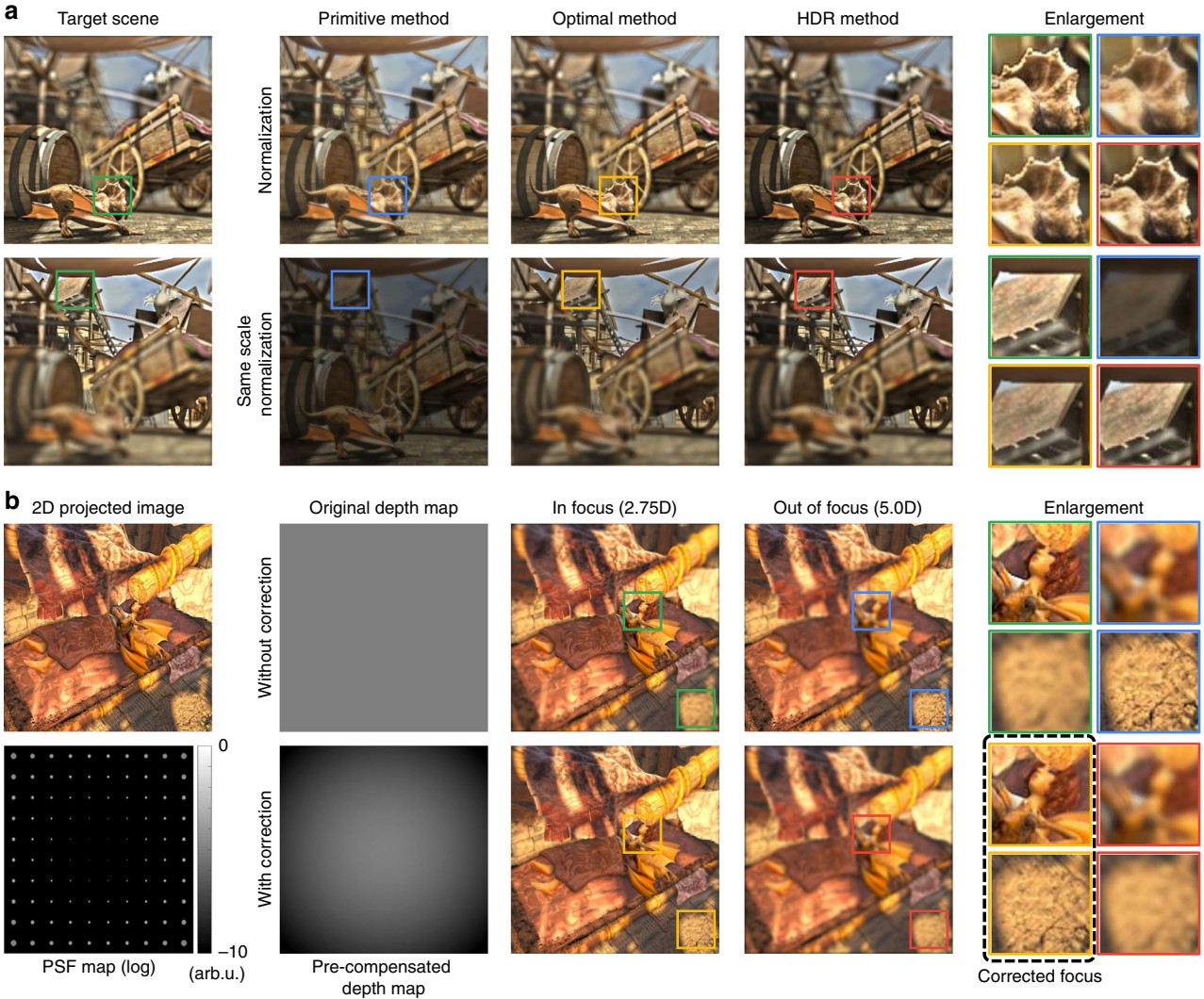

**Fig. 7** Description of advanced applications of tomographic displays. We simulate tomographic displays based on wave optics, where the number of focal planes is 80, and the offset luminance is set to 0.05. **a** We compare the HDR method with the primitive and optimal methods. The HDR method shows the most convincing performance in terms of preserving the original contrast. Additional comparisons with experiments are available in Supplementary Note 6. **b** The capability of aberration correction is demonstrated. A 3D content with a constant depth is employed for intuitive demonstration. We assume that the display system has a specific optical aberration (i.e., field curvature) as described in the PSF map. The optical aberration is represented by using Seidel coefficient: $W_{220} = 10\lambda$. This aberration could be compensated by a modification of the depth map

following least-squares problem:

$$
\min \sum_{k=1}^{p^2} \left\| \mathbf{V}_k - \sum_{j=1}^{m} \mathbf{P}_{(k,j)} \mathbf{L}_j \right\| = \min \sum_{k=1}^{p^2} \left\| \mathbf{V}_k - \sum_{j=1}^{m} \mathbf{P}_{(k,j)} (\mathbf{B}_j \odot \mathbf{D}) \right\|, \quad (2)
$$
$$
B_j = [\mathbf{b}_j, \mathbf{b}_j, \mathbf{b}_j],
$$

where $\mathbf{P}_{(k,j)}$ denotes the projection matrix between the $j$th layer and the $k$th perspective view image. Note that the optimization parameters are $\mathbf{b}_k$ and $\mathbf{D}$.

The least-squares problem has specific constraints, due to the characteristics of the FSAB: $\mathbf{b}_k$ should be binary (0 or 1) and $\mathbf{D}$ should be bounded between 0 and 1. The binary constraint makes the least-squares problem NP-hard. We decided to find the optimized parameters by solving the relaxation of the problem with the carefully designed initial condition. The initial condition is established based on the optical structure of the tomographic displays. For details on the solver, initial condition, convergence graph, and optimal solution, see Supplementary Note 2.

**Real-time rendering of binary image sequence.** In tomographic displays, each pixel of the display module would be illuminated during a specific moment. Note that the illumination time for each pixel is important, because it determines the brightness, contrast, and resolution of the display. When the illumination time is too short, the tomographic display may suffer from the limited brightness and contrast. If the illumination time is too long, the duplicated images of pixels are

extended along the depth direction, which may degrade the resolution and obscure the focus cues.

The depth fidelity of each pixel in a tomographic display is spatially invariant and independently determined by binary sequences. Thus, we can extend the solution of a simplified optimization problem for 3D scene representation. The simplified problem is to find optimal binary sequences that reconstruct a point light source at a specific depth. We suppose a point light source located at the desired depth of $z_d$ and derive its incoherent point spread function (PSF)[39] according to the focal plane depths of $z_1^s, \ldots, z_m^s$. The set of PSF, $h(z_1^s, z_d), \ldots, h(z_m^s, z_d)$, is considered as the ground truth, which is desired to be reconstructed by the tomographic display. Note that we will derive an optimal illumination strategy in the Fourier domain, because human visual characteristics are represented by a function related to spatial frequency.

For the cost function $J$, we employ the visual difference between the ground truth of PSF and the intensity profiles reconstructed by the tomographic display. By minimizing the cost function via numerical approaches, we can find the optimal illumination strategy. The cost function $J$ is given by

$$
J = \sum_{i=1}^{m} \| H(z_i^s, z_d) - P(z_i^s) \|^2, \quad (3)
$$

where $H(z_i^s, z_d)$ denotes the Fourier transform of the PSF when the depth of the focal and image planes are $z_i^s$ and $z_d$, respectively. $P(z_i^s)$ is the Fourier transform of the reconstructed intensity profile at the depth of $z_i$. The reconstructed intensity

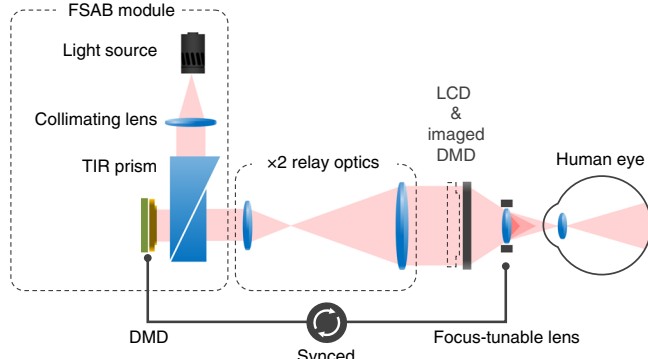

**Fig. 8** Implementation of benchtop tomographic near-eye displays. Relay optics may create a magnified real image of the DMD screen at the LCD plane, which is referred to as the FSAB. The DMD is synchronized with the tunable lens (60 Hz) to provide depth information. The DMD projection system could be replaced by an LED array backlight[44], which has advantages in the form factor. Note that the eyepiece is not illustrated in this figure. Detailed specifications and photographs of the prototype are presented in Supplementary Notes 5 and 6

profile is determined by the sum of the PSF values corresponding to each focal plane as described by the following equation:

$$P(z_i^s) = \frac{1}{A} \sum_{j=1}^{n} b_j H(z_i^s, z_j^t), \; A = \sum_{j=1}^{n} b_j, \quad (4)$$

where $z_1^t, \dots, z_n^t$ is the depth of the focal plane, whose pixel is illuminated by the corresponding backlight pixel. The on/off state (0/1) of the backlight pixel is determined by a binary sequence of $b = \{b_1, \dots, b_m\}$ referred to as the illumination strategy. $A$ is the sum of the binary sequence $b$, which is a normalization constant that determines the illumination time during a single cycle as the $A/m$ frame. Accordingly, we may formulate a least-squares problem to find the optimal illumination strategy as follows:

$$\min_b \; \sum_{i=1}^{m} \left\| H(z_i^s, z_d) - \frac{1}{A} \sum_{k=1}^{n} b_k H(z_i^s, z_k^t) \right\|^2. \quad (5)$$

As the PSF from each focal plane is incoherently merged via addition without destructive interference, the solution of the least-squares problem is trivial. The optimal illumination strategy is to turn on each pixel for as short as possible. The normalization constant $A$ should be minimized to 1, which means that the maximum brightness of the reconstructed scenes is degraded by $1/m$ times. It could be considered a trade-off between the number of focal planes and the brightness of the reconstructed scenes. When supporting 80 focal planes, tomographic displays may suffer from a low brightness that is $1/80$ times that of ordinary display panels. The low brightness could be a barrier expanding the dynamic range of the displays.

To secure a certain level of brightness, we may suppose the lower bound for the normalization constant, $A$. The lower bound could be considered as a constraint for the least-squares problem. The cost function is modified to

$$J = \sum_{i=1}^{m} \| H(z_i^s, z_d) - P(z_i^s) \|^2 + \gamma(A_{\text{low}} - A), \quad (6)$$

where $A_{\text{low}}$ is the lower bound to secure minimum brightness, and $\gamma$ is a regularization parameter. For instance, $A_{\text{low}}$ is set to $0.625 \, m$ when the desired brightness is higher than 5/8 times that of ordinary 2D displays.

In Eq. 4, we suppose an ideal environment where the backlight pixel could be completely zero. In a practical sense, however, there could be a leakage caused by a backlight diffuser, which is referred to as offset luminance. In other words, each pixel of the focal plane is illuminated by a constant brightness, even if the corresponding backlight pixel is turned off. For consideration of this artifact, we may modify the equation for the reconstructed intensity profiles as follows:

$$P(z_i^s) = \frac{1}{A} \sum_{j=1}^{n} (b_j + c) H(z_i^s, z_j^t), \; A = \sum_{j=1}^{n} (b_j + c), \quad (7)$$

where $c$ is referred to as the offset luminance determined by the amount of light source leakage.

The last point to determine an optimal illumination strategy is the consideration of human visual characteristics. First, the response of the human visual system to accommodation varies according to the spatial frequencies[40,41]. We can consider this characteristic by appending the contrast sensitivity model $V$

($f$) introduced by Mantiuk et al.[32]. The comprehensive cost function is given by

$$\min_b \; J, \; J = \sum_{i=1}^{m} V(f) \| H(f; z_i^s, z_d) - P(f; z_i^s) \|^2 + \gamma[A_{\text{low}} - A]. \quad (8)$$

Second, we note that the eye lens of the human visual system has some aberration[41] that influences the PSF. To derive a more accurate PSF, we apply ordinary human eye models demonstrated by Zernike polynomials[42].

To find the optimal binary sequence $b$ that minimizes the cost function given by Eq. 8, we employ the genetic algorithm[43]. Note that the $b_1, \dots, b_m$ should be 0 or 1, which is another constraint for the optimization problem. In the genetic algorithm, the maximum number of generations and the population size are set to 1000. Figure 6 shows the optimization results according to the degree of offset luminance and the desired brightness. In this simulation, we suppose that tomographic displays support the depth range between 5.5D (0.18 m) and 0.0D (infinity) with 160 focal planes. The spatial frequencies of the reconstructed scenes are bounded between 0 and 10 cpd. The diameter of the human pupil is assumed to be 6 mm. The optimization took ~ 30 s.

Using the optimal illumination strategy determined by solving the least-squares problem, we can render a binary image sequence with a depth map of a volumetric scene. When the offset luminance of the display system is estimated as 0.025, the optimal illumination time is 8/480 s. The illumination time corresponds to the eight focal planes in our prototypes, which means that the FSAB illuminates each display pixel by eight times during a single cycle. Finally, the binary image sequence could be derived via simple image processing. An image processing tool to render plane images is implemented using CUDA, which can be operated in real time. It takes 6 ms to render 80 binary images of 450 × 450 resolution, where we use a 3.6-GHz 64-bit Intel Core i7 CPU with 8 GB of RAM and GTX 970.

## Data availability

The data that support the plots within this paper and other findings of this study are available from the corresponding author upon reasonable request. The source data underlying Figs. 1 and 7 and Supplementary Figs. 1, 13, 14, 16–18, and 20 can be accessed at http://sintel.is.tue.mpg.de/. The source data underlying Figs. 3 and 5 and Supplementary Figs. 4 and 7–9 are provided as a Source Data File (https://doi.org/10.6084/m9.figshare.8011277).

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

## Acknowledgements
This work was supported by Institute for Information & Communications Technology Promotion (IITP) grant funded by the Korea government (MSIT) (No. 2017-0-00787, Development of vision assistant HMD and contents for the legally blind and low visions).

## Author contributions
S.L. conceived the concept of tomographic displays, occlusion blending, optimal back-light illumination strategy, and optical design of the benchtop prototype for tomographic near-eye displays. Y.J. implemented the benchtop prototype, executed the experiment, and simulated the field curvature with S.L. D.Y. synchronized focus-tunable lens with fast spatially adjustable backlight, and implemented a draft prototype of tomographic displays that employs an LED array backlight. J.C. performed the optical simulations with S.L. and analyzed the results. D.L. verified the optimization algorithm and optical simulation. B.L. initiated and supervised the projects. All of the authors discussed the results and reviewed the paper.

## Additional information

**Competing interests:** The authors declare no competing interests.

