## [Peer Review File · Nature Communications]

Reviewers' comments:

Reviewer #1 (Remarks to the Author):

This is a fairly well-written paper. Although recently there are quite a few papers in the subject area of 3D displays with both of parallax and depth cues, the current work stands out in demonstrating one type of 3D displays with continuous focal planes based on one focus-tunable lens and a fast adjustable backlight sub-system. Thus I recommend this work can be published in Nature Communications after a few modifications that would help the readers.

1. There may be some misunderstandings on the definition of "eye box". The eye box is the 3D zone in which eyes can observe the virtual images clearly. In Line 101, "the eye-box of 7.5mm" should be the exit-pupil of 7.5mm. It is better to give the value of eye relief as well.

2. In the experimental set-up, one camera was employed to simulate our eyes. It is better to give the detail of this camera, such as entrance pupil and F number.

Reviewer #2 (Remarks to the Author):

The paper proposes a new type of multifocal displays. The proposed display is composed of a focus-tunable lens, a liquid-crystal display (LCD), and a fast spatially adjustable backlight (FSAB) implemented by mapping a digital micro-mirror device (DMD) onto the LCD. By synchronizing the DMD with the focal length configuration of the focus-tunable lens, each pixel on the LCD can be displayed at certain virtual depths. Since the pixels on the DMD can be turned on and off swiftly, the proposed display can generate 80 focal planes with 60 frames per second.

1) The composition of the FSAB and the LCD is clever and innovative. This enables each pixel can be flashed within a short amount of time, which bypasses the need of a high-speed display and reduces the blur caused by changing the focus of the tunable lens.

2) While the proposed display can generate 80 focal planes in a range of 5.5 diopters, these focal planes are not independent since the content at the same pixel on two different focal planes cannot have arbitrarily different intensities --- this is a key limitation. The paper proposes to apply optimization to design the contents on the focal planes so that the limitations on brightness and occlusion can be alleviated, but the effectiveness of such optimization is suspect.

3) The paper is missing key experiments to support the claim that 80 focal planes are precisely displayed at 80 different depths. Since the focus-tunable lens is in the oscillatory mode, the blur induced by the change in focus should limit the number of total focal planes that can be displayed accurately. This essentially reduces the number of independent focal planes that can be displayed. Further, there is no discussion on the repeatability of the focal plane locations and their stability with time.

4) The paper claims that the proposed method expands the eyebox. In my understanding, the claim is true only when the whole scene is continuous in depth. In Section A, the paper uses a slanted plane to show that reducing focal-plane separation reduces the overlapping/separation regions. However, natural scenes contain lots of depth discontinuities, which cannot be handled by reducing focal-plane separation. These depth discontinuities will still cause focal planes to overlap and separate when the eye is not at the designed position even when the display has a very small focal-plane separation. This effect can be clearly seen from the attached supplemental video. So I would expect these claims to be toned down or at least the limitations to be pointed out.

5) In line 39, the paper says the resolution of the display is 450×450 . However, both the DMD and the LCD has resolution greater than 1920×1080 . I tried to search in the paper but cannot find the reason of the mismatch, or maybe I missed it. Where does the drop of resolution come from?

6) I also wonder the effectiveness of binary blending. In the case when two focal planes are assigned half of the pixel intensity, if the eye box shifts, the combined pixel intensity at the angle will be different from the original pixel intensity. So is such blending helpful? The paper uses Figure 4 to show the effect of binary blending on depth discontinuity. However, the choice of scene, namely a slanted plane, represents an easy case. I would expect to see scenes with general cases of depth discontinuity which consists of different extents of depth gaps. Could the paper comment and explore such cases? In addition, the reference display used in Figure 4 is also a multifocal display. I think using rendered real images as a reference would be more meaningful since that is the target in these cases.

8) Why Equation (3) is solved in Fourier domain? From my understanding, it can also be solved in spatial domain. What is the benefit over solving in spatial domain?

7) There are two concurrent works that are closely related to the ideas in this paper. Both tackle AR/VR type displays capable of high focal plane density. Might be worth discussing them given the similarities.

Chang, Jen-Hao Rick, B. V. K. Kumar, and Aswin C. Sankaranarayanan. "Towards Multifocal Displays with Dense Focal Stacks." arXiv preprint arXiv:1805.10664 (2018).

Rathinavel, K., Wang, H., Blate, A., & Fuchs, H. "An Extended Depth-of-Field Volumetric Near-Eye Augmented Reality Display". IEEE transactions on visualization and computer graphics (2018).

Other comments

- Line 159: only the binary constraint makes the problem NP-hard.

- The line under Equation (3), it should be "depth of accommodation and image planes".

Reviewer #3 (Remarks to the Author):

This paper describes work on a so-called "TomoReal" Display. This is a type of 3D display, and is one of many in the literature aimed at improved the realism and comfort of 3D displays. The authors make a number of claims of the advantages of such a display - the main one being associated with the accommodation vergence conflict - but they also describe advantages of high dynamic range and and the ability to remove aberrations.

In my opinion, this paper does not warrant publication in Nature Comms. The paper does contain some interesting ideas but they are rather incremental. Indeed if I were reviewing this for one of the key subject specific journals I think the case for publication is limited and the claims overblown.

So - the paper is a clear modification of the technique where a display is synchronised to a switchable lens. Other papers are correctly cited in the paper - but the novelty here is to also have a fast switchable backlight behind the display which allows the speed (and the number of depth planes) to be increased. This is an interesting idea - but a relatively minor deviation form an already established idea.

I recommend that the authors re-write the paper and town done the claims (I see no reason to give this the name TomoReal), and re-submit it to a regular optics or display journal.

Authors' reply to reviewers' comments on manuscript

Dear reviewers,

First, we appreciate valuable and detailed reviews. We are pleased to receive such constructive comments. It would be a great honor to apply reviewers' counsels to our work. We revised the paper considering all the points that the reviewers commented.

Responses to reviewer #1

[Comment 1]

“There may be some misunderstandings on the definition of "eye box". The eye box is the 3D zone in which eyes can observe the virtual images clearly. In Line 101, "the eye-box of 7.5mm" should be the exit-pupil of 7.5mm. It is better to give the value of eye relief as well.”

[Reply] Thank you for precise comment about the definition of eye-box. We revised the manuscript and provided more information of eye relief.

In revised manuscript, we adjusted “eye-box” to “exit-pupil” if the rectification is required.

On page 3, the 1st paragraph in Section “**Results**”:

- We implement a prototype for tomographic displays, which we call TomoReal. TomoReal is designed as a near-eye display for virtual reality. The eyepiece secures enough eye relief (50 mm) for observer while maintaining the optimal field of view that can be achieved with the focus-tunable lens.

[Comment 2]

“In the experimental set-up, one camera was employed to simulate our eyes. It is better to give the detail of this camera, such as entrance pupil and F number.”

[Reply] We apologize for obscure exposition of detail specification of camera. We revised the manuscript to give clear information of experimental set-up.

On page 10, the 2nd paragraph in Section “**Method**”:

- ... Using this lens, TomoReal may have depth of field between 10.5D (0.095m) and 0D (infinity) while TomoReal only supports 5.5D due to the camera lens specifications (TUSS LYM1614, focal length of 16mm, F number 1.4, entrance pupil size of 35.5mm).

Responses to reviewer #2

[Comment 1]

“while the proposed display can generate 80 focal planes in a range of 5.5 diopters, these focal planes are not independent since the content at the same pixel on two different focal planes cannot have arbitrarily different intensities --- this is a key limitation. The paper proposes to apply optimization to design the contents on the focal planes so that the limitations on brightness and occlusion can be alleviated, but the effectiveness of such optimization is suspect.”

[Reply] As reviewer #2 commented, proposed displays have a limitation that it is difficult to represent independent focal plane images. Due to the limitation, our system could suffer from the artifacts at the occlusion boundary. According to the previous research [Narain et al. 2015], independent modulation of focal plane images is important to reconstruct the occlusion boundary.

Occlusion blending, however, is introduced to verify that it is also feasible to alleviate the artifacts via appropriate operation of backlight module. In occlusion blending, each pixel is illuminated by multiple times (0-8) for representation of single voxel. Then, the intensity of voxels could be modulated according to the illumination time. Note that intensity modulation of RGB colors is also feasible if color sequential illumination is applied.

In a point of view, the principle of occlusion blending is similar with that of currently published works [Chang et al., 2018, Rathinavel et al., 2018]. These works use HDR illumination source that is synchronized with DMD. By modulating illumination time as well as intensity of illumination source, they generate full-color voxels with 8-bit gradation. In the same way, our prototype has some flexibility to modulate focal plane images even though voxels from the identical pixel of LC panel still have high correlation.

For efficient illumination time modulation, we conceived optimization problem and introduced a solver of the problem. The optimization solver could find agreed point between the fidelity of reconstructed 3D scenes and representation of occlusion boundaries. As shown in simulation and experimental results, the solver has verified the feasibility to represent occlusion boundary via our prototype.

[Comment 2]

“There are two concurrent works that are closely related to the ideas in this paper. Both tackle AR/VR type displays capable of high focal plane density. Might be worth discussing them given the similarities.

Chang, Jen-Hao Rick, B. V. K. Kumar, and Aswin C. Sankaranarayanan. "Towards Multifocal Displays with Dense Focal Stacks." arXiv preprint arXiv:1805.10664 (2018).

Rathinavel, K., Wang, H., Blate, A., & Fuchs, H. "An Extended Depth-of-Field Volumetric Near-Eye Augmented Reality Display". IEEE transactions on visualization and computer graphics (2018)."

[Reply] Thank you for suggestion about related works that are recently published. We have closely reviewed these papers and found some similarities and differences. We are pleased to discuss these ideas in the revised manuscript.

Chang et al. [2018] and Rathinavel et al. [2018] reported multi-focal displays that also employed focus-tunable lens and a display module that can be updated at high speed (660Hz). The display module was implemented by using a DMD and a high dynamic range illumination source (HDR LED). The DMD and the HDR LED are synchronized for direct digital synthesis [Lincoln et al. 2017] so that the DMD could express full color images with 8 bit gradation using 24 binary frames. Accordingly, they could implement a full color

display module that can be updated at 660 Hz where the DMD operates at 16 kHz. If this module is synchronized with focus-tunable optics, 11 focal plane images could be generated at different depths with 0.6D spacing. Since each focal plane image is independently updated, the prototypes may have some advantages in representation of occlusion boundaries.

In these works, however, we may observe the decline of the resolution and the contrast since adjacent focal planes are separated by 0.6D. When linear or optimal blending is applied for multi-plane displays, high frequency information of 3D scenes is distorted or blurred as demonstrated in Narain et al.'s paper [2015]. This limitation was also discussed in Fig. 4 and Fig. S.3 of the revised manuscript. On the other hand, our prototype with dense focal planes has advantages in the reconstruction of high resolution images.

Note that direct digital synthesis using 24 binary frames also involves the decline of the resolution as well as the contrast. Since each focal plane is represented by 24 binary images extended along the specific dioptric range (0.6D), the resolution and the contrast are declined due to the optical blur. We believe that this optical blur degrades the reliability of the occlusion boundary representation. However, the feasibility of occlusion boundary representation was not demonstrated via experiment or precise simulation.

In addition, DMD is essential element because their prototypes require binary displays that support high frame rate (> 16 kHz). In order to generate 11 focal plane images with full color, 8-bit gradation in color, and 60 FPS, binary display image should be updated at 15.8 kHz. It is quite demanding specifications for displaying a video, interaction with users, and implementation of wearable device.

Compared to these works, our prototype does not necessarily employ DMD projection system. We can use a LED array instead of the DMD, which enables us to implement wearable prototypes as shown in the revised manuscript. Note that the LED array is not compatible with the direct digital synthesis because it supports much less resolution (8 by 8) and frame rate (< 1 kHz) compared to DMD. In our prototype, additional display panel complement much higher resolution (491 dpi) as well as color expression.

The application of LC panel provides more versatility of tomographic displays in various aspects. First, tomographic displays could alleviate the specific optical aberration (i.e. curvature of field) with the modification of the rendering. Second, high dynamic range (HDR) display is also a feasible application of tomographic displays.

On page 3, the last paragraph in Section “**Tomographic Displays**”:

- Recently, Chang et al. and Rathinavel et al. reported multi-focal displays that also employed focus-tunable lens and a display module. The display module was implemented by using a DMD and a high dynamic range illumination source (HDR LED). The DMD and the HDR LED are synchronized for direct digital synthesis so that the DMD could express full color images with 8 bit gradation using 24 binary frames. Accordingly, they could implement a full color display module that can be updated at 660 Hz where the DMD operates at 16 kHz. If this module is synchronized with focus-tunable optics, 11 focal plane images could be generated at different depths with 0.6D spacing. Since each focal plane image is independently updated, the prototypes may have some advantages in representation of occlusion boundaries.

In these works, however, we may observe the decline of the resolution and the contrast. When linear or optimal blending is applied for multi-plane displays with 0.6D layer spacing, high frequency information of 3D scenes is distorted or blurred as demonstrated in Narain et al.'s paper [2015]. Second, direct digital synthesis using 24 binary frames involves further decline of the resolution and the contrast. Since each focal plane is represented by 24 binary images extended along the specific dioptric range (0.6D), the resolution and the contrast are declined due to the optical blur. On the other hand, our prototype with dense focal planes has advantages in the reconstruction of high resolution images.

In addition, DMD is essential element because their prototypes require binary displays that support high

frame rate (> 16 kHz). It is quite demanding specifications for displaying a video, interaction with users, and implementation of wearable device. Compared to these works, our prototype does not necessarily employ DMD projection system. We can use a LED array instead of the DMD, which enables us to implement wearable prototypes. Note that the LED array is not compatible with the direct digital synthesis because it supports much less resolution (8 by 8) and frame rate (1 kHz) compared to DMD. In our prototype, additional display panel complement much higher resolution (491 dpi) as well as color expression.

On page 10 of Supplementary Material, Figure S. 10:

Figure S.10. Photographs of TomoReal. On the left hand side, we present TomoReal that employs DMD projection system as FSAB. On the right hand side, we introduce a wearable prototype of TomoReal that uses a LED array (Adafruit 64 by 32 LED Matrix) as FSAB. The LED array could support 8 by 8 resolution and update a binary image at 480Hz, which can generate 8 tomographic layers.

[Comment 3]

“The paper is missing a key experiments to support the claim that 80 focal planes are precisely displayed at 80 different depths. Since the focus tunable lens is in the oscillatory mode, the blur induced by the change in focus should limits the number of total focal planes that can be displayed accurately. This essentially reduces the number of independent focal planes that can be displayed. Further, there is no discussion on to repeatability of the focal plane locations and their stability with time.”

[Reply] We apologize to reviewers for insufficient demonstration of the focal plane numbers. Additional experimental results are supplemented to show the 80 focal plane reconstruction. For more precise experiment, we employed a different camera lens of higher numerical aperture. These results support the claim of Figure S.6, which is originally intended to verify reconstruction of 80 focal planes. We also captured an oscilloscope image and supplemented discussion on to repeatability and stability of our prototypes.

On page 5 of Supplementary Material, in Section “**Reconstruction of 80 Focal Planes**”:

- As demonstrated in the manuscript, TomoReal supports 80 focal planes between 18cm (5.5D) and infinity (0.0D). In other words, each focal plane of TomoReal is separated by 0.07D. **The 0.07D separation is too narrow to be observed by a c-mounted camera lens because the depth of field of the lens is larger than 0.07D. Thus, we employ a DSLR camera (Canon EOS 5D) with a 50mm lens (Canon 50mm 1:1.4 EF), which has higher numerical aperture. Although this camera could not capture full field of view of prototype, it is appropriate to estimate the depth of focal planes.**

Figure S.7 demonstrates the experimental results. In the experiment, TomoReal reconstructs 80 points that are floated at different depths between 0.0D and 5.5D. For estimation of the each point depth, point spread

function of TomoReal is estimated according to the depth of image plane as shown in the first row of Fig. S.7. Note that the focal length of focus-tunable lens is set as constant when we derive corresponding point spread function. Using this point spread function, we predict the depth of 80 reconstructed points. As shown in the results, TomoReal reconstructs focal planes at desired depths with the convincing accuracy.

Although TomoReal shows convincing reliability of focal plane reconstruction, we may observe some error of focal plane depths. The error seems to be caused by slight mismatch in synchronization of focus-tunable lens and FSAB due to the different latency of two devices. Nevertheless, NI board resolves the mismatch periodically so that the overall system does not accumulate the errors. Note that users rarely observe the error since human visual system cannot detect the high frequency vibration. In addition, we believe that this issue could be settled by introduction of feedback circuit in commercialization step.

Figure S.7. Experimental results to estimate the depth of focal planes. As shown in the first row, point spread functions of TomoReal according to the focal length of focus-tunable lens are estimated. The second row demonstrates 80 reconstructed points at different depths between 0.0D and 5.5D. We estimate the blur size of each point, which is compared with point spread functions for prediction of reconstructed depths. Note that first 5 and last 4 points have the same depths, respectively, because FSAB illuminates each point at adjacent 8 focal planes. On the right hand side, we captured oscilloscope images to show stability and synchronization between both electrical signals into DMD and focus-tunable lens.

[Comment 4]

“The paper claims that the proposed method expands the eyebox. In my understanding, the claim is true only when the whole scene is continuous in depth. In Section A, the paper uses a slanted plane to show that reducing focal-plane separation reduces the overlapping/separation regions. However, natural scenes contain lots of depth discontinuities, which cannot be handled by reducing focal-plane separation. These depth discontinuities will still cause focal planes to overlap and separate when the eye is not at the designed position even when the display has a very small focal-plane separation. This effect can be clearly seen from the attached supplemental video. So I would expect these claims to be toned down or at least the limitations to be pointed out.”

[Reply] We agree with reviewer’s comment that eye-box could be limited if 3D contents have large depth discontinuities. Our original intention is to demonstrate that separation or overlap of adjacent focal planes is barely observed within the eye-box when the number of focal planes is larger than 80. We revised the misleading manuscript and toned down our claim as follows.

On page 4, in Section “**Determination of the Number of Focal Planes**”:

- ~~Eye-Box Expansion:~~ **Determination of the Number of Focal Planes**

It has been an interesting topic to debate how many focal planes are necessary to reconstruct a volumetric object without noticeable artificial effects. According to perceptual studies, human visual system has limited depth of field (0.15D) so that users may not recognize the discrete structure of multi-plane system with layer spacing smaller than 0.15D. In practical uses, however, the layer spacing of 0.15D is not enough to conceal the discrete structure **because of separation or overlap between adjacent focal plane images by the pupil movement**. We note that more focal planes enable display system **to have more tolerance for the pupil movement**, which provides the intuition to determine the focal plane number of TomoReal.

Multi-focal system usually assumes that an observer's pupil is fixed at a specific point so that multi-plane images are properly synthesized at the retina. When the pupil is dislocated from the specific point, multi-plane images could be misaligned by relative disparity. This misalignment may cause the resolution loss or distortion of accommodation cues. Accordingly, misalignment issue restricts the eye-box of entire system as a fixed point. As limited eye-box is not desired for near-eye displays, some advanced methods have been presented to expand the limited eye-box with or without a gaze-tracking system.

In tomographic displays, the misalignment of adjacent focal planes is alleviated when the number of focal planes is increased. As human visual system has the resolution limitation of 30 cycle per degree (cpd), **tomographic displays with dense focal planes** have some tolerance for pupil movement. When layer spacing of 0.058D is supported, users may not observe artifacts **caused by misalignment of adjacent focal planes within 10mm**. Our prototype of 0.07D layer spacing **has tolerance for pupil movement up to 7.5mm**, which is larger than the exit-pupil determined by optical system. Detailed analysis and experimental results are demonstrated in Supplementary Material.

In summary, dense focal planes improve the tolerance for the pupil movement. However, it could not solve all challenging issues involved in representation of 3D scenes. If 3D scenes have large depth discontinuities, our prototype is also vulnerable to the pupil movement. For mitigation of the artifacts caused by depth discontinuities, we should adopt computational methods rather than optical solution. First, we may use approximated depth map with smooth variation, which was employed by Matsuda et al.. We can also apply blending methods that optimize focal plane images to have more tolerance for depth discontinuities, which will be demonstrated in the following section.

On page 1 of Supplementary Material, in Section “**Alignment of Tomographic Displays**”:

- **Misalignment in Tomographic Displays**

Figure S.1 illustrates how we analyze misalignment of focal planes in tomographic displays. When the pupil of observer is dislocated from the desired position, focal plane images are misaligned due to the disparity. Note that this misalignment is usually natural effect referred to as motion parallax. However, it causes artifacts including separation and overlap when multi-plane images reconstruct a volumetric object such as a slanted bar that extends along the several layers. In that case, the misalignment make observers to feel that volumetric objects have discontinuous pattern.

Since human visual system has a resolution limit about 30cpd, observer may not recognize the separation or overlap of adjacent plane images caused by pupil movement when the misalignment is small enough. In order to ensure that observer could not see this misalignment, layer spacing should be narrow enough. We can derive a relationship between the **degree of misalignment** and layer spacing as follows. ... According to this equation, **tomographic displays secure expanded tolerant region of 10mm when layer spacing is narrower than 0.058D. Our prototype with 0.075D spacing has tolerance for pupil movement of 7.5mm.**

[Comment 5]

*“In line 39, the paper says the resolution of the display is 450*450. However, both the DMD and the LCD has resolution greater than 1920*1080. I tried to search in the paper but cannot find the reason of the mismatch, or maybe I missed it. Where does the drop of resolution come from?”*

[Reply] We apologize to reviewers for insufficient demonstration of the display resolution. As reviewer #2 commented, the DMD and the LCD have greater resolution than 450 by 450. However, the focus-tunable lens has limited aperture (~10 mm) so that it is hard to use the full area of the LCD panel. In this study, we employed small part (~23 mm by 23 mm) of LCD panel that covers 450 by 450 pixels. We revised manuscript for clear demonstration as follows.

On page 11, 2nd paragraph in Section “**Methods**”:

- For liquid crystal panel, we used Topfoison TF60010A model whose backlight module is eliminated. This panel may support high resolution images of 491 DPI. **Note that partial area (~23mm×23mm) of LC panel is employed as display module due to the limited numerical aperture of focus-tunable lens and eyepiece.** Focus-tunable lens (EL10-30-TC-VIS-12D) of Optotune is selected for the focus-tunable optics, which provides the wide focus tuning range between 8.3D to 20D at the real-time operation (60Hz). Using this lens, TomoReal may have depth of field between 10.5D (0.095m) and 0D (infinity) while TomoReal only supports 5.5D due to the camera lens specifications (TUSS LYM1614, focal length of 16mm, f/1.6). Note that we append the eyepiece module for TomoReal in order to retain the enough eye relief (50mm) for photographs. The eyepiece module consists of two identical camera lenses (Canon EF 50mm f/1.8 STM) for the 4f relay system.

[Comment 6]

“I also wonder the effectiveness of binary blending. In the case when two focal planes are assigned half of the pixel intensity, if the eye box shifts, the combined pixel intensity at the angle will be different from the original pixel intensity. So is such blending helpful? (continuous) The paper uses Figure 4 to show the effect of binary blending on depth discontinuity. However, the choice of scene, namely a slanted plane, represents an easy case. I would expect to see scenes with general cases of depth discontinuity which consists of different extents of depth gaps. Could the paper comment and explore such cases? In addition, the reference display used in Figure 4 is also a multifocal display. I think using rendered real images as a reference would be more meaningful since that is the target in these cases.”

[Reply] As reviewer #2 commented, the combined pixel intensity could be different from the original intensity due to the eye box shift. For discussion of the validity of binary blending, we suppose two pixels (A and B) that are reconstructed by adjacent focal planes as shown in following figure.

According to the eye-box shift, synthesized image could be distorted as illustrated in the figure. If the two pixels A and B are not correlated so that they have independent colours, binary blending cannot improve display performance. In practical use, however, those pixels are expected to have some correlation since they are close each other in terms of the depth as well as the angle.

If A and B are originated from the identical continuous surface, binary blending may alleviate the artifacts caused by separation or overlap of adjacent focal planes. If binary blending is not applied, combined pixel intensity is zero (separation) or 2 times larger than original intensity (overlap). When binary blending is applied, distorted combined pixel intensity is 0.5 times smaller (separation) or 1.5 times larger (overlap) than original intensity.

For demonstration of the advantages of binary blending, we supplemented simulation results using complex 3D scenes instead of a simple continuous surface. More simulation results are supplemented as follows.

On page 2 of Supplementary Material, the last paragraph in Section “**Details in Analysis of Depth Discontinuities**”:

- Based on the analysis, we simulate how pupil movement affects the synthesis of focal plane images. Figure S.2 demonstrates that pupil movement may degrade the image fidelity of tomographic displays. We can observe the pupil movement distortion that could not be noticed in the experiment (Fig. S.1) due to the resolution limit. In addition, we recognize that binary blending enables tomographic displays to have some tolerance for the pupil movement.

For more intuitive comparison and evaluation of tomographic displays with binary blending, simulation

results using a 3D scene are presented in Fig. S.3. As shown in the results, tomographic displays could provide higher resolution retinal images with high frequencies. Binary blending alleviates the artifacts caused by misalignment of adjacent focal planes. Although binary blending could not resolve the occlusion boundary issues, it would be a convincing and efficient solution for real-time operation.

Figure S.2. The effect of pupil movement for synthesis of focal plane images. We suppose a slanted bar that covers depth range of 0.6D with a sinusoidal texture of 20 cpd. When the slanted bar is reconstructed by multi-plane or tomographic displays, the sinusoidal texture of 20 cpd is observed as described in composite retinal images. Each row illustrates how retinal images are affected by the pupil movement (-1mm, 0mm, and 1mm). Note that we consider multi-plane display with dense focal planes (0.075D spacing) as a reference system. Tomographic displays with binary blending show the most convincing performance to provide finite retinal images of high fidelity.

Figure S.3. Comparison and evaluation of binary blending using a 3D scene. The 3D scene has the depth range of 5.5D (18cm-infinity), which is observed by a 4mm pupil that focuses on the depth of 3.7D. On the right hand side, enlargements of images demonstrate the advantages of binary blending as well as tomographic displays. First, multi-plane displays could not provide high frequency information of 3D

scenes. Second, binary blending alleviates the artifacts that occur when pupil is shifted by 1.5mm from the desired position.

[Comment 7]

“Why Equation (3) is solved in Fourier domain? From my understanding, it can also be solved in spatial domain. What is the benefit over solving in spatial domain?”

[Reply] As reviewer #2 commented, Equation (3) could be solved in spatial domain if we do not consider human visual characteristics. In order to reflect the human visual characteristics (contrast sensitivity function) related to spatial frequency, we solved least squares problem in Fourier domain. This approach was consulted in a previous research presented by Narain et al. [2015]. For more intuitive exposition, we revised manuscript as follows.

On page 7, 2nd paragraph in Section “**Fundamental Architecture for Optimization**”:

- The depth fidelity of each pixel in tomographic displays is spatially invariant and independently determined by binary sequences. ..., is considered as the ground truth, which is desired to be reconstructed by tomographic displays. **Note that we will derive optimal illumination strategy in Fourier domain because human visual characteristics are represented by a function related to spatial frequency.**

[Comment 8]

“Line 159: only the binary constraint makes the problem NP-hard.

The line under Equation (3), it should be “depth of accommodation and image planes”.

[Reply] Thank you for precise review and detailed review on the paper. We revised the manuscript and corrected the errors.

On page 6, 4th paragraph in Section “**Occlusion Blending**”:

- The least squares problem has specific constraints due to the characteristics of the FSAB: b_k should be binary (0 or 1) and D should be bounded between 0 and 1. **The binary constraint makes** the least squares problem NP-hard.

On page 8, 3rd paragraph in Section “**Fundamental Architecture for Optimization**”:

- For the cost function J , we employ the visual difference between the ground truth of PSF and intensity profiles reconstructed by tomographic displays. ... where $H(z_i^s, z_d)$ denotes Fourier transform of PSF when **the depth of accommodation and image planes** are z_i^s and z_d , respectively.

Responses to reviewer #3

[Comment 1]

“In my opinion, this paper does not warrant publication in Nature Comms. The paper does contain some interesting ideas but they are rather incremental. Indeed if I were reviewing this for one of the key subject specific journals I think the case for publication is limited and the claims overblown.

So - the paper is a clear modification of the technique where a display is synchronised to a switchable lens. Other papers are correctly cited in the paper - but the novelty here is to also have a fast switchable backlight behind the display which allows the speed (and the number of depth planes) to be increased. This is an interesting idea - but a relatively minor deviation from an already established idea.

I recommend that the authors re-write the paper and tone down the claims (I see no reason to give this the name TomoReal), and re-submit it to a regular optics or display journal.”

[Reply] As reviewer #3 commented, there have been various related researches. We conceived proposed displays with small idea, which is combination of tunable lens, fast switchable backlight, and “LC panel”. Although it seems not really novel and innovative, we claim that most related researches have not employed the “LC panel”. The application of LC panel provides more versatility and flexibility to use different type of fast switchable backlight. For instance, we could apply LED array backlight and implement a wearable prototype.

The introduction of occlusion blending is also important and valuable contribution of our study. We have proposed a draft solver for binary least squares problem (NP-hard), which is actively studied in related fields. Although our solver is so primitive that precise initial condition is required for convergence, it is first time to introduce optimization solver for binary least squares problem for multi-focal displays. We believe that our work could inspire a new research to improve optimization solver by applying the idea of non-binary discrete tomography introduced by Zisler et al. [2016].

We also emphasize that our research could have large impact in various display fields. First, we believe that it is the first time to present head-mounted display system that presents several focal planes (> 8 planes). Second, any display system that employs temporal multiplexing methods could consider the combination of display panel and switchable backlight. Third, we could explore human visual characteristics related vergence-accommodation conflict by using our prototypes.

On page 10 of Supplementary Material, Figure S. 10:

Figure S.10. Photographs of TomoReal. On the left hand side, we present TomoReal that employs DMD projection system as FSAB. On the right hand side, we introduce a wearable prototype of TomoReal that uses a LED array (Adafruit 64 by 32 LED Matrix) as FSAB. The LED array could support 8 by 8 resolution and update a binary image at 480Hz, which can generate 8 tomographic layers.

Thank you.

Reviewer #1:

Remarks to the Author:

The authors have replied all the comments from the reviewers comprehensively. The authors have presented a new type of new type of real-3D near-eye displays with well performance. This work would be very much of interest to researchers of near-eye displays, 3D displays and imaging processing.

Reviewer #2:

Remarks to the Author:

The revised manuscript addresses many of the points raised in my review for the previous round. The effort on part of the authors is appreciated.

On the positive side, the revised paper handles my comments on eyebox, 80 depth planes, and has toned down some claims and clarified others.

However two points remain --- both very easy to address but the revised manuscript has serious flaws because of not handling them.

1) The paper's discussion of Chang et al and Rathinavel et al. is misleading and simply put, wrong. As such this part has to be rewritten from scratch.

First, Chang et al. shows a capability of 40 depth planes over a 4D range. hence, their depth spacing is $0.16D$ --- not $0.6D$ as claimed in the paper.

Second, Rathinavel's design allows for 280 depth planes over a range of $6.7D$ --- a depth resolution of $0.025D/\text{depth plane}$ --- again, --- not $0.6D$ as claimed in the paper. This capability is even better than that of the submission.

So the para in the paper about these techniques having $0.6D$ depth resolution is plainly wrong.

Fourth, the designs of Chang et al. and Rathinavel et al. are not very different from what is proposed. All of them use the liquid lens in a resonant mode and so are able to get a large number of depths --- 40 in the case of Chang, 80 for the submission, and 280 for Rathinavel. Each design compromises in a different place.

Chang allows for full control over the pattern at each bit plane, but has to multiplex to do color --- either by using 3DMDs or by losing bit depth.

Rathinavel provides 280 depth planes --- but each of them is a binary bit-depth.

Finally, tomoreal uses two displays --- but there is no independent control of the pattern shown at same pixel at different bit planes.

Each has their merits and demerits.

Finally, the paper mentions Chang and Rathinavel -- but does not cite them in the references. Please fix this.

2) The response to my "Comment #1" which points out that intensity of a pixel at different depth planes is fixed due to the nature of the hardware design. This is a limitation of the hardware design since the color display is way slower than the DMD. To provide a complete description of the pros and cons of the display, it would be best to simply acknowledge this limitation.

Reviewer #3:

Remarks to the Author:

I see that I am out of line with the other two referees. However, my fundamental point about this

paper remains. This paper oversells itself and claims to explain a fundamentally new technique - whereas it is a relatively minor modification of an existing idea. The modifications to the paper have not address this point.

Authors' reply to comments of editor and reviewers on manuscript

Dear reviewers,

First, we appreciate valuable and detailed reviews. We are pleased to receive such constructive comments. It would be a great honor to apply reviewers' counsels to our work. We revised the paper considering all the points that the reviewers commented.

Responses to reviewer #1

[Comment 1]

“The authors have replied all the comments from the reviewers comprehensively. The authors have presented a new type of new type of real-3D near-eye displays with well performance. This work would be very much of interest to researchers of near-eye displays, 3D displays and imaging processing.”

[Reply] Thank you for your precise comments in the previous round.

Responses to reviewer #2

[Comment 1]

“The paper’s discussion of Chang et al and Rathinavel et al. is misleading and simply put, wrong. As such this part has to be rewritten from scratch.

First, Chang et al. shows a capability of 40 depth planes over a 4D range. Hence, their depth spacing is 0.16D --- not 0.6D as claimed in the paper.

Second, Rathinavel’s design allows for 280 depth planes over a range of 6.7D --- a depth resolution of 0.025D/depth plane --- again, --- not 0.6D as claimed in the paper. This capability is even better than that of the submission.

So the para in the paper about these techniques have 0.6D depth resolution is plainly wrong.

Fourth, the designs of Chang et al. and Rathinavel al. are not very different from what is proposed. All of them use the liquid lens in a resonant mode and so are able to get a large number of depths --- 40 in the case of Chang, 80 for the submission, and 280 for Rathinavel. Each design compromises in a different place.

Chang allows for full control over the pattern at each bit plane, but has to multiplex to do color --- either by using 3DMDs are losing bit depth.

Rathinavel provides 280 depth planes --- but each of them is a binary bit-depth.

Finally, tomoreal uses two displays --- but there is no independent control of the pattern shown at same pixel at different bit planes.

Each have their merits and demerits.

Finally, the paper mentions Chang and Rathinavel -- but does not cite them in the references. Pl fix this.”

[Reply] Thank you for precise correction about the descriptions of related works (Chang et al. and Rathinavel et al.). After reviewing the reviewer #2’s comment, we figured out that our description could be controversial because we have arbitrarily interpreted the specifications of related prototypes. In the revised manuscript, we tried to compare related prototypes without any interpretation in the specifications. We revised the description of the related works as follows.

On page 2, the 2nd paragraph in Section “**Multi-Plane and Volumetric Displays**”:

- Recently, Chang et al. and Rathinavel et al. reported volumetric displays that have a large number of focal depths, 40 and 280, respectively. The display module was implemented by using a DMD and a high dynamic range illumination source using light emitting diode (HDR LED). The DMD and the HDR LED are synchronized for direct digital synthesis that decomposes colour images into binary image sequence. Applying direct digital synthesis, the prototypes of two research groups result in efficient binary representation of 3D imagery. However, both prototypes have some disadvantages in frame rate or bit-depth. Chang et al.'s prototype lacks of bit-depth and frame rate as it reproduces 8-bit gray images updated at 40FPS. Rathinavel et al.'s prototype also suffers from the limited bit-depth since each focal plane image has a binary bit-depth.

In order to compare these prototypes, we defined two evaluation criteria, referred to as upper bound amplitude in signal and bit-depth. The upper bound amplitude is Fourier coefficient of retinal images, and bit-depth denotes the degree of freedom to modulate signal intensity. We believe the evaluation criteria could provide valuable intuition to understand advantages and disadvantages of the candidates for volumetric displays. In manuscript, we compare candidates for near-eye displays with focus cues by using the criteria.

On page 5, the 1st paragraph in Section “**Evaluation of Display Capability**”:

- In order to assess tomographic displays, we define two evaluation criteria: upper bound amplitude and bit-depth. The upper bound amplitude is Fourier coefficient of synthesized images, and bit-depth denotes the degree of freedom to modulate pixel brightness. These two criteria give insight into the contrast, resolution limit, and color depth of tomographic displays. Figure 4 demonstrates the analysis of upper bound amplitude and bit-depth supported by tomographic displays. Other state-of-the-art prototypes of related researches are also assessed to show the competitiveness of tomographic displays. Among the candidates, tomographic displays have the most promising potential for representation of high frequency information as well as high dynamic range images. The detailed definition and derivation of the upper bound amplitude and bit-depth can be consulted in Supplementary Material.

On page 5, Figure 4:

Figure 4. Analysis of upper bound amplitude and bit-depth according to the spatial frequency and accommodation depth. The simulation result compares three prototypes of Rathinavel et al., Chang et al., and us. Rathinavel et al.'s prototype supports limited depth of field and bit-depth at high spatial frequency, and Chang et al.'s prototype lacks bit-depth for full color representation. On the other hand, tomographic displays show tolerant performance reliable of spatial frequency and accommodation depth.

The definition and derivation of these two evaluation criteria are described in Supplementary Material.

On page 1 of Supplementary Material, Section “**Derivation of Upper bound Amplitude and Bit-Depth**”:

- In order to analyze contrast and resolution limit of volumetric displays and tomographic displays, we define maximum and upper bound amplitudes which are given by

$$MA(v, z_a) = \max \left(\sum_{j \in (z_a - z_j) < \frac{1}{wv}} b_j L_j(v) H(v, z_a; z_j) \right) = L_m(v) \sum_{j \in (z_a - z_j) < \frac{1}{wv}} b_j H(v, z_a; z_j), \quad (S. 1)$$

$$MA(v, z_a) \leq UA(v, z_a) = G \sum_{j \in (z_a - z_j) < \frac{1}{wv}} b_j H(v, z_a; z_j),$$

where $MA(v, z_a)$ and $UA(v, z_a)$ are maximum and upper bound amplitudes, respectively, which are referred to as Fourier coefficients (spatial frequency: v) of retinal images when focal depth of human eye is z_a . The Fourier coefficients can be derived from the sum of transfer function $H(v, z_a; z_j)$ of focal plane images, $b_j L_j(v)$. The luminance of backlight, b_j , may vary according to focal planes when direct digital synthesis is applied. Considering the finite-aperture version of the depth of field, we select adequate focal plane images sufficiently near from the focal depth of human eye. The selected focal plane images support maximum displayable frequency higher than the interested spatial frequency, v . Note that all focal plane images have the identical maximum Fourier coefficient, $L_m(v)$, whose upper bound is supposed to be a constant, G (Parseval's theorem). Finally, we derive the upper bound amplitude that is proportional to the sum of transfer function.

As described in Eq. S. 1, the upper bound amplitude is a function related to the spatial frequency and the focal depth. The upper bound amplitude can be used as evaluation criteria of the contrast and resolution limit. First, we suppose that the upper bound amplitude of zero frequency is normalization constant. The normalization constant is referred to as the upper bound luminance of display systems. Second, we calculate the ratio of the upper bound amplitude to the normalization constant according to the spatial frequency. The ratio indicates how bright images of corresponding frequency can be reconstructed through display system. If the ratio is close to zero, the brightness of the image could be too dim to be observed. Ideal display system would have constant upper bound amplitude, 1, regardless of accommodation depth as well as spatial frequency.

The bit-depth is also an important factor to evaluate display performance. For representation of full color 8 bit images, display system should have 24 bit-depth. In tomographic displays, at least 24 bit-depth is supported by the additional display panel. In fact, tomographic displays would have higher bit-depth because we can modulate the signal intensity by modulating illumination time of the backlight. For instance, the signal intensity of zero spatial frequency could have approximately 79 times more depths that corresponds to the increase of 6.3 bit-depth. Note that the bit-depth increase becomes slow at the high spatial frequency because only a few adjacent focal planes are coupled. The number of coupled focal planes is derived based on the upper bounds on the spatial frequency.

Using the same methodology, we can also analyze the bit-depth of related prototypes as shown in the manuscript. For the prototype of Rathinavel et al., each focal plane has 1 bit-depth while combination of adjacent focal planes gives exponential variation due to the binary illumination. The exponential increase in bit-depth continues until at least 24 focal planes are coupled. For the prototype of Chang et al., each focal plane has 8 bit-depth. In this prototype, the increase tendency in bit-depth is similar with that of tomographic displays.

In order to support the analysis using the upper bound amplitude and the bit-depth, we demonstrate retinal image simulation results where we consider voxel-oriented display system of Rathinavel et al.'s work. In addition, multi-plane displays with 11 and 80 focal planes are referenced for comparison with tomographic displays. We replaced the previous figure (Fig. 5) with the simulation results that are derived by updated optimization solver for occlusion blending. Note that the updated solver shows much better performance than the previous one.

On page 5, the 2nd paragraph in Section “**Evaluation of Display Capability**”:

- For more quantitative evaluation of tomographic displays, we also conducted retinal image simulation to see how focal plane images are synthesized. We compare tomographic displays with 80-plane displays to investigate the drawback that comes from the slow frame rate of the display panel (60FPS) in tomographic displays. Contrary to tomographic displays, each focal plane image of 80-plane displays can be independently determined according to the blending method such as linear blending or optimal blending. In this simulation, we assume that all systems have resolution limit of 20 cpd where the horizontal field of view is set to 10°. We employ several visual metrics including peak signal to noise ratio (PSNR), image quality factor (Q), and HDR-VDP-2 that estimates the probability for users to detect artifacts. Figure 5 demonstrates the simulation results that verify the validity of using the display panel to increase the number of focal planes and bit-depth simultaneously. As shown in the results, tomographic displays show comparable display performance to the 80-plane displays where each focal plane is determined independently.

On page 6, Figure 5:

Figure 5. Quantitative evaluation of tomographic displays. A volumetric scene (Source image courtesy: “SimplePoly Urban”, www.cgtrader.com) extends along the depth range between 1.0D and 3.4D. In the bottom row, we demonstrate probability map of detection for visual difference between reconstructed scenes and the ground truth. Each pixel value indicates the weighted average of probability over all accommodation depths of observers. Average of PSNR (Avg. PSNR) is derived from the weighted sum of errors between the ground truth and synthesized retinal images. The weight is estimated by the reciprocal of the optical blur kernel size. Average of Q (Avg. Q) is the mean values of all focal stack images' Q. Average of probability map (Avg. P) is the mean value of all pixel's detection probability. Tomographic displays show the similar display performance with 80-plane displays in terms of Avg. Q, and Avg. P. In Supplementary Material, we present focal plane images for each display system and additional comparison results from other related systems.

On page 10 of Supplementary Material, the 1st paragraph in Section “Additional Results for Evaluation of Tomographic Displays”:

- In this section, we will demonstrate the competitive performance tomographic displays compared to following display systems: multi-plane displays of 11 layers with 0.6D spacing and voxel-oriented multi-plane displays of 280 layers. Note that multi-plane displays of 11 layers are representative system for conventional multi-plane displays where four planes are separated by 0.6D. For comparison, we apply the identical quantitative analysis introduced in the manuscript where all systems have resolution limit of 20 cpd where the horizontal field of view is set to 10°. Retinal images are derived by synthesizing 7 by 7 sampled multi-view images on the pupil plane. The pupil size of human eyes is set to 6mm. As demonstrated in Figure S. 8, tomographic displays enable users to observe the most accurate scenes with minimized errors. Tomographic displays show the highest values of average peak signal noise ratio (Avg. PSNR), average quality factor (Avg. Q), and average probability to detect artifacts (Avg. P). Figure S. 9 illustrates the focal plane images that are used for the comparison.

On page 10 of Supplementary Material, Figure S. 8:

Figure 5. Quantitative evaluation of tomographic displays via comparison from other candidate systems. As shown in the results, tomographic displays enable users to observe the most accurate retinal images. Note that the performance of Rathinavel et al.'s system is degraded by the chromatic distortion and artifacts at occlusion boundaries. If an adequate optimization algorithm is introduced in the future work, the performance is expected to be enhanced.

On page 12 of Supplementary Material, Figure S. 9:

Figure 5. Quantitative evaluation of tomographic displays via comparison from other candidate systems. As shown in the results, tomographic displays enable users to observe the most accurate retinal images. Note that the performance of Rathinavel et al.'s system is degraded by the chromatic distortion and artifacts at occlusion boundaries. If an adequate optimization algorithm is introduced in the future work, the performance is expected to be enhanced.

Along with the analysis of upper bound amplitude and bit-depth, we supplemented the discussion about the necessity of black frames. We found that human observers may notice artificial effects when a moving scene is reconstructed without insertion of black frames. Revised manuscript includes the brief exposition of this issue involved by multi-plane displays using the temporal multiplexing method. The detailed description of the artificial effects is presented with simulation results in Supplementary Material.

On page 4, the 3rd paragraph in Section “**Tomographic Near-Eye Displays for Virtual Reality**”:

- Along with the promising display performance demonstrated in experiment, tomographic near-eye displays have two more advantages. First, it is capable of inserting black frames without the decrease in the focal plane number since our prototype does not necessarily increase the number of focal planes to the utmost limit of DMD system (about 280 planes). We note that black frames contribute to mitigation of the undesired artifacts when a video is played. Without black frames, users may observe irregular stripe patterns caused by simultaneous observation of focal plane image stacks in adjacent frames. Second, we can use a LED array instead of the DMD in order to implement wearable prototypes. The LED array supports much less resolution (8 by 8) and frame rate (<1 kHz) compared to the DMD. In tomographic displays, however, the additional display panel complements the limitation of the LED array by supporting much higher resolution (491 dpi) as well as 24 bit-depth colours. Supplementary Material presents detailed demonstrations of the necessity of back frames and wearable tomographic displays using a LED array.

On page 1 of Supplementary Material, Section “**Necessity of Black Frames**”:

- For multiple focal plane reconstruction, the focus-tunable lens is operated by periodic signal such as sinusoidal or triangle wave. We may suppose two methods for arrangement of focal plane images in the periodic cycles where every focal depth is scanned twice within a single period. The first one is to display half of focal plane images during the forward cycle, and display the other half of focal plane images during the backward cycle as shown in Fig. S.1. This method allows tomographic displays to increase the number of focal planes up to 280 focal planes, which is identical to that of Ratinavel et al.'s prototype. The second one is to employ only half of periodic cycle for representation of volumetric scenes while the other half cycle remains as a black frame. Although the second method loses frame rate, the black frame insertion is important for this type of displays when moving scenes are played.

Tomographic displays synthesize multiple focal plane images via temporal multiplexing. If a single cycle for the multiplexing takes a time less than 1/60 seconds, human observers recognize the focal plane images as a volumetric scene. However, human observers could be more sensitive and recognize the artificial effects of tomographic displays when they are exposed to a moving scene without black frames. We provide a brief exposition of this phenomenon in Fig. S.1. Since human observer likely recognizes adjacent frame images simultaneously, even a slight movement of volumetric scenes becomes noticeable artifacts when focal plane images are synthesized without black frames. The synthesized images have stripe patterns where focal plane images of adjacent frames are not fitted well. On the other hands, the stripe patterns are alleviated when black frames are inserted.

On page 2 of Supplementary Material, Figure S.1:

Figure S.1. Retinal image simulation of moving scenes when black frames are inserted or not. Note that we simulate various conditions to verify that the stripe patterns are recognized regardless of the exposure times when black frames are not inserted. The results show that black frame insertion is efficient solution to alleviate the stripe pattern artifacts. With back frames, tomographic displays provide users with much more natural motion blur effect that is similar with conventional 2D displays.

[Comment 2]

“The response to my “Comment #1” which points out that intensity of a pixel at different depth planes is fixed due to the nature of the hardware design. This is a limitation of the hardware design since the color display is way slower than the DMD. To provide a complete description of the pros and cons of the display, it would be best to simply acknowledge this limitation.”

[Reply] As reviewer #2 commented, tomographic displays have a limitation in the representation of independent focal plane images. In the revised manuscript, we toned down of our claim and acknowledged the limitation with the discussion of future work.

On page 5, the 1st paragraph in Section “**Alleviation of Depth Discontinuity Artifacts: Occlusion Blending**”:

- Although tomographic displays have various advantages as demonstrated in the previous section, it could be premature to consider tomographic displays as the most promising system for virtual reality. Since focal plane images are merged via addition, tomographic displays could be vulnerable for the depth discontinuity artifacts at occlusion boundaries. Without an adequate solution, synthesis of focal plane images seems to be artificial when the depth discontinuities are significant. In previous researches related to multi-plane displays, it was verified that linear or optimal blending could alleviate the depth discontinuities. Unfortunately, tomographic displays could not apply those blending methods directly because all focal plane images are correlated with each other. Each focal plane image of tomographic displays could not be independently determined because FSAB divides a constant RGB image into multiple focal plane images. Therefore, we need to conceive an alternative blending method to minimize depth discontinuities.

On page 7, the 1st paragraph in Section “**Discussion**”:

- In this study, we have implemented and explored the prototype for tomographic near-eye displays considered as one of the most promising systems to reconstruct 3D objects with continuous focus cues. There are some interesting issues and challenges for improvement in the performance of tomographic displays, which could be valuable discussion for future works. First, tomographic displays have a limitation that it is difficult to represent independent focal plane images. Due to the limitation, tomographic displays could suffer from the artifacts at the occlusion boundary. Although we have demonstrated that tomographic displays are also able to mitigate the artifacts via occlusion blending, further researches should be followed for the real-time operation. We believe that the real-time operation could be feasible by using a convolutional neural network. Second, enhancement in the brightness is important if we aim to stack large amount of focal planes. We could set low boundary A_{low} to achieve higher brightness or optimize optical path of system to minimize the loss of illumination.

Responses to reviewer #3

[Comment 1]

“I see that I am out of line with the other two referees. However, my fundamental point about this paper remains. This paper oversells itself and claims to explain a fundamentally new technique - whereas it is a relatively minor modification of an existing idea. The modifications to the paper have not address this point.”

[Reply] In the revised manuscript, we tried to tone down our claim and replaced “TomoReal” with a generic term such as tomographic near-eye displays or implemented prototype. In addition, we supplemented additional subsection “Related Work” to acknowledge the previous methodologies that inspire us to conceive tomographic displays.

- Please understand that all of the details of above revision are highlighted in the response of reviewer #2’s comments to avoid duplication.

Additional Revision

[Topic 1] Enhancement of optimization solver for occlusion blending.

[Explanation] During the revision periods, we found a way to enhance the optimization solver for occlusion blending. The updated solver has additional step called regularization of backlight energy. Using this updated solver, we achieved much enhanced display performance of tomographic near-eye displays. The enhanced performance was verified via experiment as well as simulation.

On page 4 of Supplementary Material, the 2nd paragraph in Section “Algorithm to Solve NP-hard Problem”:

- In this study, we solved the optimization problem on the left side. In order to solve this problem using SART, we ignore the binary constraint and update B_k for some iteration. Then, we calculate the backlight energy distribution of display pixels by adding up B_k . The backlight energy distribution indicates that illumination time of each display pixels. Using the distribution map as a milestone, we convert B_k to an appropriate set of integers so that the changes of backlight energy distribution as well as errors between B_k and updated B_k are minimized. We call this process as regularization of backlight sequences, which is one of the most important step for convergence of this algorithm. After the regularization step, we go back to the first step to update B_k and repeat this procedure for some iterations. Finally, we may find approximated solution for the NP-hard problem.

On page 5, Figure 3:

Figure 3. Simulation and experimental results to demonstrate validity of occlusion blending. A volumetric scene (Source image courtesy: “Interior Scene”, www.cgtrader.com) extends along the depth range between 0.0D and 4.0D. As shown in the results, occlusion blending enables tomographic displays to represent volumetric scenes without noticeable artifacts even at occlusion boundary. Additional results with experiment are available in Supplementary Material.

Figure S.4. On the left-hand side, convergence graph of proposed algorithm is demonstrated. The graph shows our algorithm is converged on an optimal point. Note that the first blue circle indicates the error of the initial condition. On the right-hand side, focal plane images are presented according to the blending methods. Note that we visualize binary image sequences for FSAB by using a merged single image. At the bottom row, experimental results are demonstrated according to the blending methods of tomographic displays. Occlusion blending shows enhanced performance to represent occlusion boundary.

[Topic 2] Rearrangement of contents for better legibility.

[Explanation] After experiencing two valuable revision processes, the contents of our paper becomes much more abundant and precise. In addition, display and optimization performance of tomographic near-eye displays are further enhanced compared to the previous manuscript of the first submission. On the other hand, we found that our paper is gradually heavier to understand and some contents in the papers are no longer important. Thus, we feel the necessity to revise the arrangement of overall composition through the papers for better legibility.

- Section “Determination of the Number of Focal Planes” was transferred to Supplementary Material (Section A.3).
- Section “Binary Blending” was transferred to Supplementary Material (Section C).
- Detailed derivation of least squares problem for occlusion blending in Section “Alleviation of Depth Discontinuity Artifacts: Occlusion Blending” was transferred to Section Method “Least Squares Problem for Occlusion Blending”
- Detailed discussion about the illumination strategy in Section “Illumination Strategy for Real-time Operation” was transferred to Section Method “Real-time Rendering of Binary Image Sequence”
- Additional 3D contents (“Interior Scene”) for illustration of occlusion blending are used for simulation and experiment. The results using previous 3D contents are transferred to Supplementary Material
- Figure for illustration of tomographic near-eye displays (Figure 2 in the previous manuscript) is transferred to method (Figure 8).
- Figure S.6 of Supplemental Material was transferred to manuscript (Figure 5) with the appropriate revision.

Thank you.

REVIEWERS' COMMENTS:

Reviewer #2 (Remarks to the Author):

The revision addresses my comments adequately.